# Nuclear membrane protein SUN2 promotes replication of flaviviruses through modulating cytoskeleton reorganization mediated by NS1

Yanxia Huang[1,2,3,7], Qinyu Peng[1,2,7], Xu Tian[1,2,7], Cancan Chen[4], Xuanfeng Zhu[1,2], Changbai Huang[1,2], Zhiting Huo[1,2], Yang Liu[5], Chao Yang[3,6] ✉, Chao Liu ✉[1,2] ✉ & Ping Zhang ✉[1,2] ✉

Cytoskeleton is extensively recruited by flaviviruses for their infection. In this study, we uncovered an essential role of a nuclear membrane protein, SAD1/UNC84 domain protein 2 (SUN2) linking cytoskeleton and nucleoskeleton in the flavivirus replication. CRISPR/Cas9-mediated knockout of SUN2, but not SUN1, significantly reduces the replication of Zika virus (ZIKV), dengue virus (DENV), and Japanese encephalitis virus (JEV). In contrast, SUN2 does not affect the infection of non-flaviviridae RNA viruses. All three regions of SUN2 are required for its proviral effect. Mechanistically, SUN2 facilitates rearrangement of cytoskeleton and formation of replication organelles induced by viral infection, and hence promotes viral RNA synthesis. SUN2 is required for the interaction between cytoskeleton actin and ZIKV nonstructural protein 1 (NS1). Expression of dominant negative Nesprin-1 and Nesprin-2, which connect SUN2 to cytoskeleton proteins, alleviates the interaction between actin and NS1 and reduces viral replication levels. In a neonatal mouse infection model, SUN2 knockout dramatically alleviates the in vivo ZIKV replication and development of neuropathology. This work elucidates that recruitment of cytoskeleton proteins by flavivirus is coordinated by nuclear membrane proteins SUN2 and Nesprins, providing evidence for a link between nuclear membrane proteins and flavivirus infection.

Flaviviruses are a family of positive-stranded RNA viruses, including yellow fever virus (YFV), dengue virus (DENV), Zika virus (ZIKV), Japanese encephalitis virus (JEV), West Nile virus (WNV), and tick-borne encephalitis virus (TBEV) virus, etc[1]. Flaviviruses are usually transmitted through arthropod vectors, and their infection might lead to severe diseases in human, posing a severe threat to global public health. For example, ZIKV is associated with neurological diseases, such as fetal microcephaly and adult Guillain-Barre syndrome[2]. DENV

[1]Key Laboratory of Tropical Diseases Control (Sun Yat-sen University), Ministry of Education, Guangzhou, China. [2]Department of Immunology and Microbiology, Zhongshan School of Medicine, Sun Yat-sen University, Guangzhou, China. [3]Department of Neurosurgery, First Affiliated Hospital of Sun Yat-sen University, Guangzhou 510080, China. [4]Department of Pathology, First Affiliated Hospital of Sun Yat-sen University, Guangzhou 510080, China. [5]School of Biomedical Sciences, Li Ka Shing Faculty of Medicine, The University of Hong Kong, Hong Kong SAR, China. [6]Department of Neurosurgery, Guangxi Hospital Division of The First Affiliated Hospital, Sun Yat-sen University, Guangxi, China. [7]These authors contributed equally: Yanxia Huang, Qinyu Peng, and Xu Tian. ✉e-mail: ychao@mail.sysu.edu.cn; liuchao9@mail.sysu.edu.cn; zhangp36@mail.sysu.edu.cn

infection affects 390 million people in tropical and subtropical regions each year, causing thousands of severe diseases including dengue hemorrhagic fever or dengue shock syndrome[3]. So far, there are no specific therapies available to treat diseases caused by flaviviruses.

Flavivirus genomic RNA encodes a polyprotein on the endoplasmic reticulum (ER) membrane, which is cleaved into three structural proteins (capsid, prM, and envelope) and seven nonstructural proteins (NS1, NS2A, NS2B, NS3, NS4A, NS4B, and NS5)[1]. A group of viral NS proteins, such as NS1, NS2A, NS4A, and NS4B, are responsible for inducing massive ER membrane remodeling and invagination, generating organelle-like compartments called replication organelles (ROs)[4–10]. ROs not only serve as sites for viral genomic RNA synthesis, but also help to sequester newly-synthesized viral dsRNA thus avoid innate immune activation. To induce ER membrane bending and ROs formation, viral proteins closely interact and recruit various cytoskeleton proteins such as actin, vimentin, and tubulin. Upon infection of flaviviruses, actin cytoskeleton network is dynamically reorganized[11–15], and microtubules and intermediate filaments are rearranged to form cage-like structures surrounding the ROs or aggregation near the nucleus[16–20]. The movement of cytoskeleton is an essential step in the flavivirus replication, as treatment with cytoskeleton inhibitors effectively abolishes formation of ROs and infection of flaviviruses[16,17].

In the cytoskeleton system, cytoplasmic skeleton is linked to nucleoskeleton through a structure called linker of nucleoskeleton and cytoskeleton (LINC) located on the nuclear envelope (NE). LINC consists of SAD1/UNC84 domain (SUN) proteins on the inner nuclear membrane (INM) and nuclear envelope spectrin repeat proteins (Nesprins) on the outer nuclear membrane (ONM)[21]. SUN proteins are central components of the LINC complex, including five members in human: SUN1, SUN2, and SUN3-5. SUN1 and SUN2 are widely expressed in various tissues and organs, while SUN3-5 are expressed in a limited amount of tissues[22]. Structurally, SUNs comprise an N-terminal nucleoplasmic domain, a transmembrane (TM) domain, a coiled-coil region located in the perinuclear space (PNS), and a SUN domain[22]. Nesprin family includes four members and a majority of them consist of an N-terminal calponin homology domain, a linker region with spectrin repeats (SRs), and a C-terminal KASH domain binding to SUN domain of SUN proteins[23]. Nesprin-1 and −2 directly bind to actin through their N-terminal domains, while Nesprin-3 and −4 lack calponin homology domain and hence associate with cytoskeleton through plectin and kinesin-1 respectively[23]. SUNs, coupling with Nesprins connect the cytoskeleton and nucleoskeleton system, and play vital roles in mechanotransduction function and cellular signaling. So far, SUNs have been reported to regulate infection of two nuclear-replicating viruses, namely human immunodeficiency virus type 1 (HIV-1)[24] and herpes simplex viruses[25]. However, whether SUNs participate in the infection of cytoplasmic-replicating viruses, particularly RNA viruses, remains unknown.

To investigate role of two widely-expressed SUNs (SUN1 and SUN2) in the replication of flaviviruses, we generated SUN knockout cell clones and mice through CRISPR/Cas9 gene editing. We found that SUN2, but not SUN1, promotes the replication of flaviviruses. SUN2 acts at the viral RNA replication step via modulating rearrangement of cytoskeleton and formation of ROs. We further revealed that SUN2, together with Nesprins, are required for the recruitment of cytoskeleton actin by ZIKV NS1. In a neonatal mice model, SUN2 knockout leads to significant reductions of in vivo viral replication levels and severity of neuropathogenesis induced by ZIKV infection.

## Results

### SUN2 plays a proviral role in the replication of flaviviruses
To explore whether SUNs participate in ZIKV infection, we generated two SUN1 and SUN2 knockout cell clones in Huh7 cells by CRISPR/Cas9 gene-editing technique (Fig. 1A, B and Supplementary Fig. 1A). Growth rates of two SUN2$^{KO}$ cells were similar to control cells (Fig. 1C). In

response to ZIKV infection, the SUN2 protein levels in Huh7 cells were gradually decreased at 24 and 48 h post-infection (h p.i.) (Fig. 1D), consistent with previous report[26]. Then, control cells transduced with empty vector (ctrl), SUN1$^{KO}$, and SUN2$^{KO}$ cells were infected with ZIKV at a multiplicity of infection (MOI) 3, and harvested at 24 h p.i. for measurement of viral replication levels. Knockout of SUN2 significantly reduced the RNA and envelope (E) protein levels, and titers of ZIKV (Fig. 1E–G). In the multi-step growth assay, SUN2 knockout led to dramatic reductions of ZIKV yields at 24, 48, and 72 h p.i. (Fig. 1H). In contrast, SUN1 depletion had no effect on ZIKV or DENV2 replication (Supplementary Fig. 1B–E).

Next, we generated SUN2$^{RES}$ cells by introducing human *SUN2* gene into the SUN2$^{KO}$ cells through lentivirus-mediated transduction. The SUN2 protein level was largely restored in the SUN2$^{RES}$ cells (Fig. 1I), and replication levels of ZIKV and DENV2, including viral protein and RNA levels, and titers, were successfully recovered (Fig. 1I–M). Similarly, the production of ZIKV and DENV2 was significantly reduced in the SUN2$^{KO}$ A549 cells, but not in the SUN1$^{KO}$ A549 cells (Supplementary Fig. 1F–J and Supplementary Fig. 2). In addition, SUN2 depletion reduced the infection of JEV (Supplementary Fig. 3).

We further tested role of SUN2 in the infection of RNA viruses belonging to other families, including three positive-strand RNA viruses (SARS-CoV-2, Semliki Forest virus SFV, and encephalomyocarditis virus EMCV) and a negative-strand RNA virus (vesicular stomatitis virus, VSV). The data showed that the SUN2 depletion had no influence on the replication of SARS-CoV-2 replicon, SFV, EMCV, and VSV (Supplementary Fig. 4A–D), indicating that SUN2 specifically promotes the replication of flaviviruses.

### All three regions of SUN2 are required for its proviral function
To test which region of SUN2 is essential for flavivirus replication, we generated stable cells expressing truncates of SUN2, including deletion of N-terminal and transmembrane (TM) domain (SUN2$^{ΔN+TM}$), deletion of coiled helix domain (SUN2$^{ΔCC}$), and deletion of SUN domain (SUN2$^{ΔSUN}$). Expression of three SUN2 truncates was validated by western blot (Fig. 2A, B). Control, SUN2$^{KO}$, SUN2$^{RES}$, and three SUN2 truncate-expressing cells were infected with mock or ZIKV, and harvested at 24 h p.i. for confocal microscopy assay using anti-SUN2 and anti-E antibodies. In the ZIKV-infected control cells, SUN2 was distributed on the nuclear membrane as predicted, and viral E protein was mainly accumulated within the perinuclear region (Fig. 2C, top panels). In the SUN2$^{KO}$ cells, SUN2 protein was not detected and viral E protein was widely dispersed in the cytoplasm. In the SUN2$^{RES}$ cells, SUN2 protein level was largely restored and distributed on the nuclear membrane; as expected, the level of ZIKV E protein was enhanced. In contrast, the SUN2 protein was predominantly located on the nuclear membrane in the SUN2$^{ΔCC}$ and SUN2$^{ΔSUN}$ cells, but in the cytoplasm of SUN2$^{ΔN+TM}$ cells (Fig. 2C). The E staining signal in all three truncate-expressing cells was significantly lower than the control cells, and was dispersed in the cytoplasm (Fig. 2C). Consistently, the viral E levels and titers in the SUN2$^{ΔN+TM}$, SUN2$^{ΔCC}$, and SUN2$^{ΔSUN}$ cells were significantly reduced (Fig. 2D, E), suggesting that all three regions are essential for SUN2 to support ZIKV replication.

### SUN2 promotes viral RNA replication and formation of replication organelles
As both RNA and protein levels of ZIKV were downregulated by SUN2 depletion, we speculated that SUN2 acts at an early step of viral infection. We first examined impact of SUN2 depletion on viral entry step. Control, SUN2$^{KO}$, and SUN2$^{RES}$ cells were inoculated with mock or ZIKV, followed by incubation either on ice for 45 min (virion binding), or at 37 °C for 30 min (virion internalization), or at 37 °C for 60 min (virion entry). Cells were harvested for total RNA extraction and qRT-PCR. The viral RNA levels in the control, SUN2$^{KO}$, and SUN2$^{RES}$ cells were

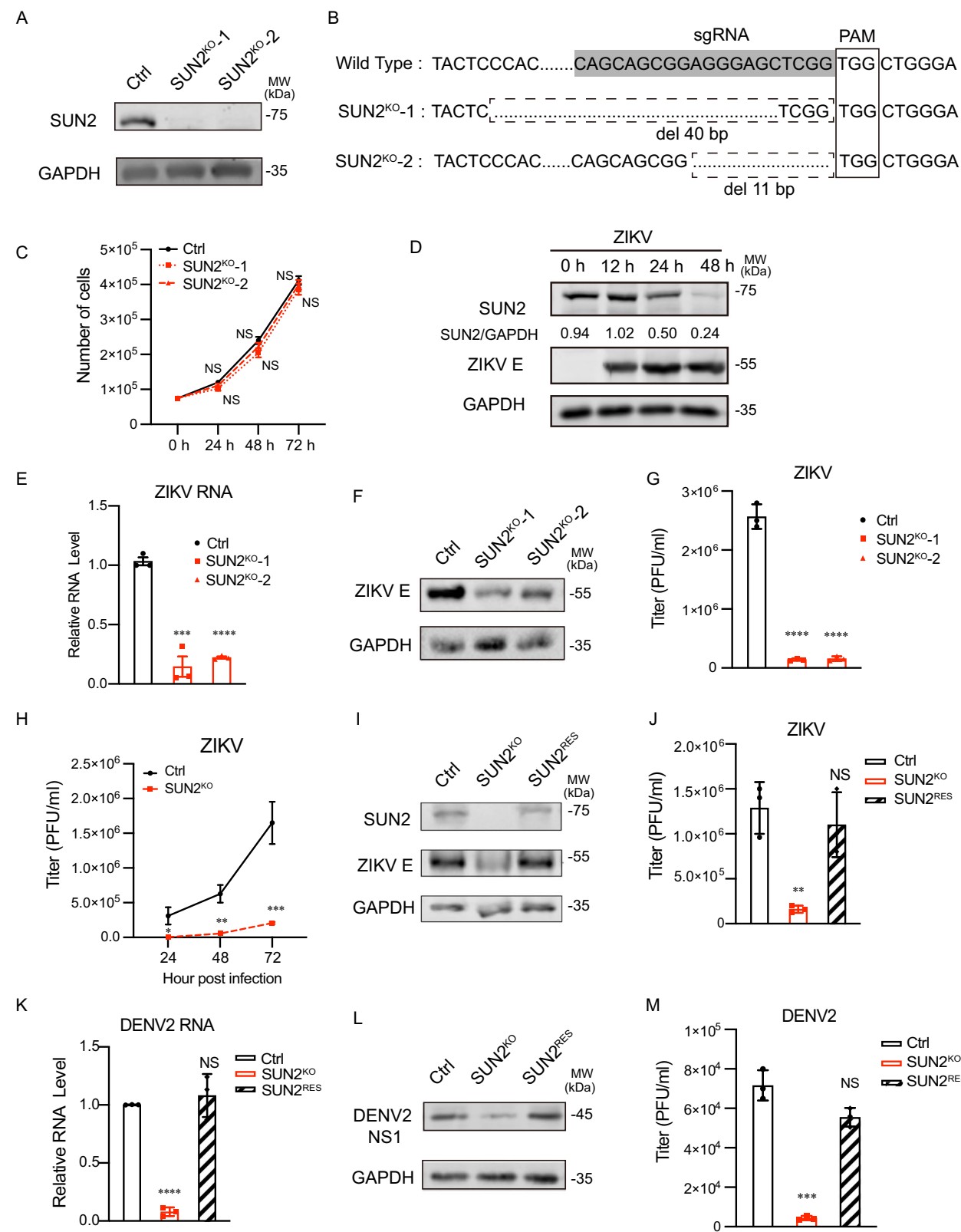

comparable in these assays (Fig. 3A), suggesting that SUN2 does not act at the viral entry step.

To examine whether protein translation or RNA synthesis of ZIKV is regulated by SUN2, we utilized a ZIKV replicon system[27]. Control, SUN2[KO], and SUN2[RES] cells were transfected with in vitro-transcribed RNAs of a wild-type ZIKV replicon (WT) or an NS5 polymerase mutant replicon (GDD) lacking RNA-dependent RNA polymerase activity. At 6,

12, 24, and 48 h post transfection (p.t.), cells were collected for luciferase assay. At 6 h p.t., the luciferase activities of WT and GDD replicon were comparable in the control, SUN2[KO], and SUN2[RES] cells, suggesting that SUN2 does not directly regulate viral protein translation. Strikingly, the luciferase activities of WT replicon in the SUN2[KO] cells were significantly lower than the control and SUN2[RES] cells at 24 and 48 h p.t. (Fig. 3B), indicating that SUN2 regulates the viral RNA

**Fig. 1 | SUN2 promotes the ZIKV and DENV2 infection in Huh7 cells. A** Knockout efficiency of two SUN2 knockout clones (SUN2$^{KO}$-1 and SUN2$^{KO}$−2) in Huh7 cells validated by western blot using anti-SUN2 antibody. GAPDH was probed as a loading control. Representative images of three independent experiments are shown. **B** Confirmation of *SUN2* gene editing by DNA sequencing. **C** Cell growth curve. Control and SUN2$^{KO}$ cell numbers were counted at 0, 24, 48, and 72 hours post inoculated. The data are shown as mean ± SD, n = 3 biologically independent samples. Statistical significance was determined using two-way ANOVA. NS, $p > 0.05$, not significant. $p$ values: $p = 0.0736$ (24 h), $p = 0.3308$ (48 h), $p = 0.4630$ (72 h). **D** Western blot. Huh7 cells were infected with ZIKV at an MOI of 3. The cells were collected at 0, 12, 24, and 48 h p.i. for western blot using antibodies against SUN2, ZIKV E, or GAPDH. Representative images of three independent experiments are shown. **E**−**G** Intracellular RNA level (**E**), E protein levels (**F**), and titers of ZIKV particles (**G**) of ZIKV-infected control and SUN2$^{KO}$ cells (MOI = 3) at 24 h p.i. were measured by qRT-PCR, western blot, and plaque assay respectively. The data are shown as mean ± SD, n = 3 biologically independent experiments. Statistical significance was determined using two-tailed unpaired *t* test. $p$ values: (**E**) $p = 0.0006$ (SUN2$^{KO}$-1), $p < 0.0001$ (SUN2$^{KO}$-2). **G** $p < 0.0001$ (SUN2$^{KO}$-1), $p < 0.0001$ (SUN2$^{KO}$-2).

**H** Multi-step growth curve of ZIKV. Control and SUN2$^{KO}$ cells were infected with ZIKV at an MOI of 0.01. The supernatants were collected at 24, 48, and 72 h p.i. for standard plaque assay. The data are shown as mean ± SD, n = 3 biologically independent experiments. Statistical significance was determined using two-way ANOVA. $p$ values: $p = 0.0491$ (24 h), $p = 0.0022$ (48 h), $p = 0.0005$ (72 h). **I**−**J** ZIKV replication levels. Control, SUN2$^{KO}$, and SUN$^{RES}$ cells were infected with ZIKV at an MOI of 3. At 24 h p.i., cells and supernatants were harvested for western blot using anti-SUN2 and anti-E antibodies or plaque assay. The data are shown as mean ± SD, n = 3 biologically independent experiments. Statistical significance was determined using two-tailed unpaired *t* test. $p$ values: $p = 0.0025$ (SUN2$^{KO}$), $p = 0.5217$ (SUN$^{RES}$). **K**−**M** DENV2 replication levels. Control, SUN2$^{KO}$, and SUN$^{RES}$ cells were infected with DENV2 at an MOI of 1. At 24 h p.i., cells were harvested for qRT-PCR (**K**) or western blot (**L**), and supernatants were collected for plaque assay (**M**). The data are shown as mean ± SD, n = 3 biologically independent experiments. Statistical significance was determined using two-tailed unpaired *t* test. $p$ values: (**K**) $p < 0.0001$ (SUN2$^{KO}$), $p = 0.8017$ (SUN$^{RES}$). (**M**) $p = 0.0001$ (SUN2$^{KO}$), $p = 0.3588$ (SUN$^{RES}$). Source data are provided as a Source Data file.

synthesis. The luciferase activities of NS5-GDD replicons were comparable among three tested cells (Fig. 3B). Moreover, the luciferase activities of WT replicon could not be restored in three SUN2 truncate expressing cells (Supplementary Fig. 6A), further demonstrating that each region of SUN2 is essential for its proviral effect.

As RNA synthesis of flaviviruses takes place in the replication organelles (ROs), we tested whether SUN2 depletion has an impact on the RO formation by transmission electron microscopy (TEM). The ZIKV-infected cells were harvested at 24 h p.i. for TEM. As expected, the ER membranes of ZIKV-infected control cells were dramatically rearranged and invaginated, with amounts of RO (indicated by yellow arrows) and viral particles (indicated by red arrows) residing in (Fig. 3C and Supplementary Fig. 5A). In contrast, the ER morphology of infected SUN2$^{KO}$ cells was barely altered, and the amounts of ROs and virions were markedly decreased (red arrows in Fig. 3C and Supplementary Fig. 5A). Likewise, the SUN2 depletion dramatically reduced the proportion of ROs and virions (Fig. 3D, E and Supplementary Fig. 5B-C).

To examine whether SUN2 affects the movement of ER membrane induced by ZIKV, we performed immunofluorescence assay (IFM) detecting two ER membrane indicators (calnexin and SEC61B) using either anti-calnexin antibody or cells expressing mCherry-SEC61B. In the mock-infected cells, the ER membranes were evenly dispersed in the cytoplasm of both control and SUN2$^{KO}$ cells were similar (Fig. 3F, G). Upon ZIKV infection, both calnexin and SEC61B were aggregated around the nuclei and co-localized with ZIKV E protein in the control cells, while SUN2 KO significantly dampened their aggregation. These data suggested that SUN2 is required for the ER membrane remodeling and formation of replication organelles induced by ZIKV.

## SUN2 modulates the rearrangement of cytoskeleton induced by ZIKV

To induce ER membrane rearrangement and formation of ROs, cytoskeleton network is reorganized upon flavivirus infection[28]. Considering SUN2 is a link protein connecting the nucleoskeleton and cytoskeleton, we hypothesized that SUN2 might direct the rearrangement of cytoskeleton induced by viral infection. To test this hypothesis, we examined subcellular distribution of three cytoskeleton proteins belonging to three classes (filament actin, microtubule tubulin, and intermediate filament vimentin) by IFM assay. In the mock-infected control and SUN2$^{KO}$ cells, the cytoplasmic localizations of F-actin, tubulin, and vimentin were all comparable (Fig. 4A−C). Upon ZIKV infection, the cytoskeleton proteins in the control cells were redistributed: a large proportion of F-actin, tubulin, and vimentin were relocalized to the perinuclear region; particularly, a cage-like structure clustering of tubulin was formed around nucleus (Fig. 4A−C). The ZIKV

E protein was mainly localized to the perinuclear region (Fig. 4A−C). In striking contrast to control cells, the distribution pattern of three skeleton proteins in the ZIKV-infected SUN2$^{KO}$ cells was different: they remained distributed all over the cytoplasm like mock-infected cells, and vimentin was even distributed to the cell periphery (Fig. 4A−C). As predicted, the viral E protein levels were lower than control cells and dispersed in the cytoplasm (Fig. 4A−C). The aggregated rates of F-actin, tubulin, and vimentin in the SUN2$^{KO}$ cells were significantly lower than the control cells (Fig. 4D−F), suggesting that SUN2 plays an essential role in the reorganization of cytoskeleton induced by ZIKV. Moreover, the *trans*-complementation of three SUN2 truncates could not restore the cytoskeleton rearrangement in the SUN2$^{KO}$ cells, and viral E protein levels (Supplementary Fig. 6B−C), indicating each region is essential for SUN2 to modulate the viral replication.

To probe whether SUN2 promotes the ZIKV infection through modulating the reorganization of cytoskeleton, we compared viral replication levels in the absence and presence of individual inhibitors of cytoskeleton proteins. Three widely-used inhibitors, including cytochalasin B (Cyt B, disrupting formation of actin polymers)[29], paclitaxel (disrupting stabilization of microtubule)[16], and acrylamide (disrupting vimentin polymerization)[30] were selected. Treatment of these inhibitors alone did not affect the cell viability of Huh7 cells (Fig. 4G). In the control cells, the ZIKV replication levels and luciferase activities of ZIKV WT replicon were remarkably reduced in the presence of individual inhibitors (Fig. 4H and Supplementary Fig. 6D), consistent with previous reports[16,29,30]. In contrast, treatment of individual inhibitors in the SUN2$^{KO}$ cells did not lead to additional reduction of viral titers (Fig. 4H, red column), suggesting the proviral effect of SUN2 relies on its regulation on cytoskeleton movement.

Next, we tested role of SUN2 in the cytoskeleton rearrangement of SFV, another positive-stranded RNA virus belonging to *Togaviridae*. SUN2 knockout did not alter the movement of F-actin or tubulin to the cell periphery induced by SFV (Supplementary Fig. 7), in keeping with the observation that SUN2 did not affect the SFV replication (Supplementary Fig. 4B). The collective data suggested that SUN2 specifically promotes the replication of flaviviruses through regulating the cytoskeleton remodeling induced by their infection.

## SUN2 facilitates the interaction between ZIKV NS1 and actin

Given reorganization of cytoskeleton induced by flavivirus infection is mainly driven by recruitment of skeleton proteins by viral nonstructural proteins[17,19], we hypothesized that SUN2 possibly modulates their interaction. To test this hypothesis, we transfected plasmids expressing ZIKV NS proteins tagged with HA at N-terminus (NS1, NS2A, NS2B, NS3, NS4A, NS4B, and NS5) into 293 T cells and detected their interactions using Co-IP assay. The interaction between NS1 and actin was readily

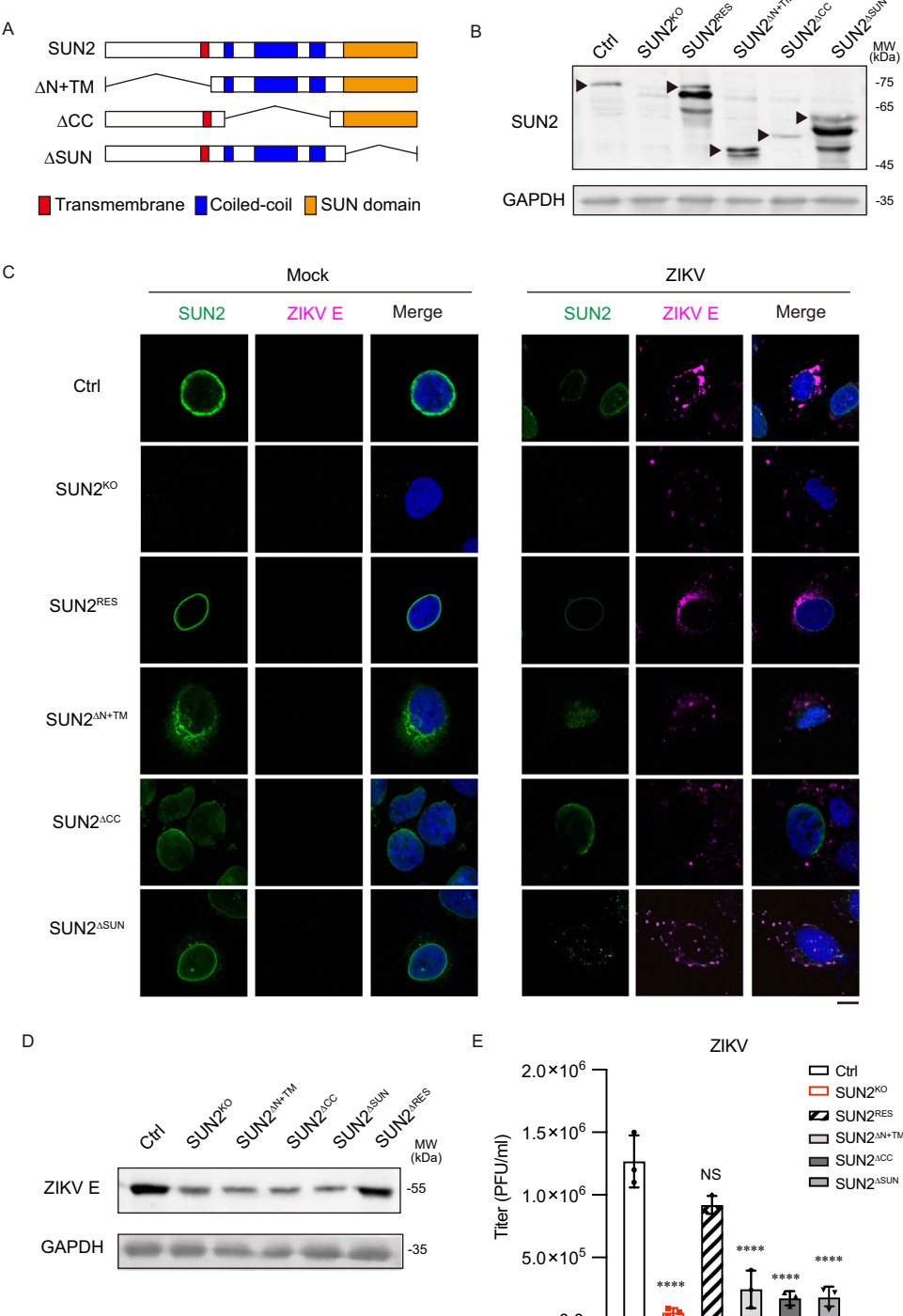

**Fig. 2 | Integrity of SUN2 is required for its proviral effect. A** Schematic representation of full-length and truncated SUN2 constructs. **B** Expression levels of three SUN2 truncates (SUN2$^{\Delta N+TM}$, SUN2$^{\Delta CC}$, SUN2$^{\Delta SUN}$) were detected by western blot. GAPDH was probed as a loading control. Representative images of three independent experiments are shown. **C** Subcellular distributions of SUN2 and ZIKV E. Control, SUN2$^{KO}$, SUN2$^{RES}$, SUN2$^{\Delta N+TM}$, SUN2$^{\Delta CC}$, and SUN2$^{\Delta SUN}$ cells were infected with ZIKV at an MOI of 8. At 24 h p.i., cells were harvested for immunostaining using anti-SUN2 (Green) and anti-ZIKV envelope (Magenta) antibodies. DAPI (Blue) stains the nuclei. Cell images were captured by Nikon C2 Confocal Microscope (Scale bars, 10 μm). Representative images of at least three independent experiments are shown. **D, E** ZIKV replication levels. Control, SUN2$^{KO}$, SUN2$^{RES}$, SUN2$^{\Delta N+TM}$, SUN2$^{\Delta CC}$, and SUN2$^{\Delta SUN}$ cells were infected with ZIKV at an MOI of 3. At 24 h p.i., cells and supernatants were harvested for western blot (**D**) and plaque assay (**E**). Representative images of three independent experiments are shown (**D**). Data are shown as mean ± SD, n = 3 biologically independent experiments (**E**). Statistical significance was determined using one-way ANOVA. $p$ values: (**E**) $p < 0.0001$ (SUN2$^{KO}$), $p = 0.1800$ (SUN2$^{RES}$), $p < 0.0001$ (SUN2$^{\Delta N+TM}$), $p < 0.0001$ (SUN2$^{\Delta CC}$), $p < 0.0001$ (SUN2$^{\Delta SUN}$). Source data are provided as a Source Data file.

detected (Fig. 5A), while other NS proteins were not or weakly associated with tubulin or vimentin (Fig. 5A and Supplementary Fig. 8A). To be pointed out, whether NS2A and NS2B directly interact with cytoskeleton proteins are unclear because they were weakly expressed.

Next, we examined whether SUN2 presence was required for the interaction between ZIKV NS1 and actin. The control and SUN2$^{KO}$ 293 T cells (knockout efficiency shown in Supplementary Fig. 8B) were transfected with plasmid expressing ZIKV HA-NS1 or HA-NS3 (negative

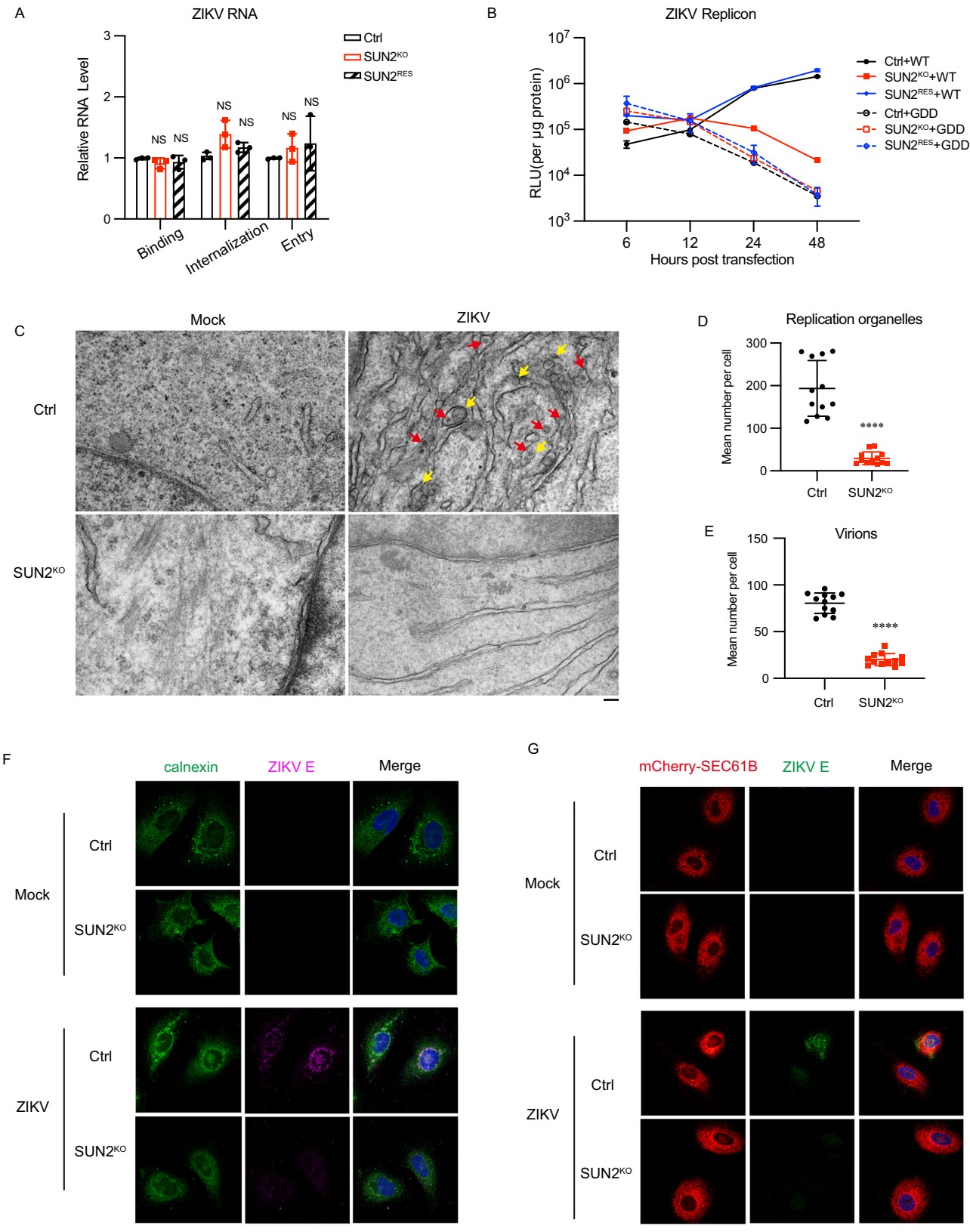

control). The SUN2 depletion largely abolished the interaction between NS1 and actin (Fig. 5B, lane 11). Reciprocal co-IP confirmed that actin was not associated with HA-NS1 in the absence of SUN2 (Fig. 5B, lanes 14 and 17). As the ZIKV protein levels in the infected control and SUN2KO cells were not identical, we were not able to directly compare the endogenous interaction between NS1 and actin.

Instead, we transfected the control and SUN2KO Huh7 cells with plasmid expressing HA-NS1, followed by ZIKV infection for 24 h, and performed co-IP assay in the context of viral infection. As shown in Fig. 5C, the SUN2 depletion severely disrupted the NS1-actin association in the ZIKV-infected context (Fig. 5C). Consistently, the IFM images showed that in the ZIKV-infected control cells, NS1 was co-localized with F-actin

**Fig. 3 | SUN2 functions at the RNA replication step of ZIKV. A** Viral entry assay. Control, SUN2$^{KO}$, and SUN2$^{RES}$ cells were inoculated with ZIKV at an MOI of 3, followed by incubation either on ice for 45 min (virion binding), or at 37 °C for 30 min (virion internalization), or at 37 °C for 60 min (virion entry). Cells were harvested for qRT-PCR. The data are shown as mean ± SD, n = 3 biologically independent experiments. Statistical significance was determined using two-way ANOVA (NS, $p > 0.05$, not significant). **B** ZIKV replicon assay. Control, SUN2$^{KO}$ cells, and SUN2$^{RES}$ cells were transfected with ZIKV WT and GDD replicon RNAs, and harvested at indicated time points for luciferase assay. The data are shown as mean ± SD, n = 3 biologically independent experiments. **C** Control and SUN2$^{KO}$ A549 cells were infected with ZIKV at an MOI of 3. At 24 h p.i., cells were fixed and

processed for TEM analysis. ROs were indicated with yellow arrows and viral particles were indicated with red arrows. Representative images of three independent experiments are shown. Scale bars, 100 nm. Quantitation of ROs (**D**), and virions (**E**) in the control and SUN2$^{KO}$ cells (n = 12 cells per condition). Graphs show the mean number per cell counted; error bars represent Mean ± SD. Statistical significance was determined using two-tailed unpaired $t$ test (****$p < 0.0001$). **F, G** IFM assay. Control and SUN2$^{KO}$ cells were infected with ZIKV at an MOI of 8. At 24 h p.i., cells were fixed and stained with anti-E (Magenta) and anti-calnexin (Green) antibodies (**F**), or anti- E (Green) antibodies (**G**). DAPI stains nucleus (Scale bars, 10 μm). Representative images of three independent experiments are shown. Source data are provided as a Source Data file.

in the perinuclear region, while in the SUN2$^{KO}$ cells, NS1 level was impaired and F-actin was not gathered around the nuclei (Fig. 5D). These collective data indicated that SUN2 is required for the NS1-actin interaction.

As ZIKV NS1 might be redistributed to different compartments such as viral replication sites upon polyprotein cleavage[31], we examined whether NS1 localizes in the ER upon ZIKV infection by IFM assay and whether SUN2 is dispensable for its movement. Control and SUN2$^{KO}$ cells expressing mCherry-SEC61B were generated by lentivirus-mediated transduction. Cells were infected with ZIKV and harvested at 24 h p.i. for IFM assay. The NS1 protein was co-localized with SEC61B in the perinuclear region of control cells (Fig. 6A), while both NS1 and SEC61B proteins showed a scattered distribution pattern in the cytoplasm of SUN2$^{KO}$ cells, indicating that SUN2 is important for NS1 to localize within ER and for ER remodeling. To test whether NS1 has an impact on rearrangement of skeleton proteins and ER, we ectopically expressed NS1 in the Huh7 cells, and performed IFM assay. Consistent with previous reports[6,32], expression of ZIKV NS1 alone in Huh7 cells was sufficient to induce remodeling of ER and actin into the perinuclear region of control cells (Fig. 6B). Depletion of SUN2 significantly reduced the rearrangement of ER and actin mediated by NS1 (Fig. 6B), demonstrating that SUN2 is involved in the NS1-mediated cytoskeleton and ER remodeling.

In addition, we tested whether SUN2 regulates the interaction between other viral proteins and skeleton proteins. Association between E and actin was not detected (Supplementary Fig. 8C), and SUN2 is dispensable for association of E and vimentin, or NS4A and vimentin (Supplementary Fig. 8D).

## SUN2 modulates the interaction of NS1 and actin through Nesprins

Considering that SUN2 is mainly located on the inner nuclear membrane, while ZIKV proteins and skeleton proteins are present in the cytoplasm, we proposed that Nesprins, which connect SUN2 and cytoskeleton proteins, might mediate the proviral function of SUN2. To test this hypothesis, we first examined whether SUN2, Nesprin-1, actin, and viral proteins are associated together by co-IP assay. The SUN2$^{RES}$ cells expressing HA-tagged SUN2 were infected with ZIKV at MOI 3. At 24 h p.i., co-IP assay was carried out using anti-HA beads. As expected, SUN2 co-precipitated with actin, Nesprin-1, and viral NS1 (Fig. 7A), implying that Nesprin-1 contributes to the regulation of SUN2 on the actin/NS1 interaction. Notably, other viral proteins including NS3, NS4A, and NS5 were also weakly detected, suggesting that these viral proteins might indirectly interact with SUN2.

To examine whether the localization of Nesprins is altered by ZIKV infection and whether SUN2 is required for their movement, we carried out IFM assay. In the mock-infected cells, Nesprin-1 was located on the nuclear membrane and dispersed in the cytoplasm regardless of SUN2 presence or absence. Interestingly, upon ZIKV infection, the cytoplasmic part of Nesprin-1 became aggregated to the perinuclear region and colocalized with ZIKV E protein in the control cells. In contrast, Nesprin-1 remained scattered in the cytoplasm of the ZIKV-infected

SUN2$^{KO}$ cells (Fig. 7B), indicating the movement of Nesprin-1 induced by ZIKV depends on the SUN2 presence.

To investigate role of Nesprins in the NS1-actin association and ZIKV replication, we constructed plasmids expressing dominant negative (DN) forms of Nesprin-1 and Nesprin-2 (only KASH domain fusing with mCherry-tag at their N-termini, designated as DN-KASH1 and DN-KASH2) (Fig. 7C). The control and SUN2$^{KO}$ cells stably expressing Nesprin KASHs were generated by lentivirus-mediated transduction. The growth rates of these cells were comparable (Fig. 7D). The IFM images showed that in the control cells, both DN-KASH1 and DN-KASH2 proteins were mainly located on the nuclear membrane or perinuclear region as they lack the cytoplasmic domains (Fig. 7E, upper panels). Interestingly, DN-KASH1 and DN-KASH2 were located not only on the nuclear membrane of SUN2$^{KO}$ cells, but also dispersed in the cytoplasm (Fig. 7E, lower panels), implying that binding to SUN2 is critical for Nesprin KASHs to be stably located on the ONM.

Then, we examined whether Nesprins regulate the NS1-actin interaction by co-IP assay. The control and SUN2$^{KO}$ cells expressing DN-KASH1 or DN-KASH2 were transfected with plasmid expressing HA-NS1, followed by ZIKV infection. At 24 h p.i., cells were harvested for co-IP assay. As shown in Fig. 7F, the interaction between NS1 and actin was significantly reduced in the control cells expressing DN-KASH1 or DN-KASH2. Expression of DN-KASH1 or DN-KASH2 in the SUN2$^{KO}$ cells further downregulated the actin/NS1 association (Fig. 7F). These data indicated that Nesprins, like SUN2, are involved in mediating the NS1-actin association.

Last, we tested the impact of DN KASHs on the ZIKV replication. Control and SUN2$^{KO}$ cells expressing mCherry, DN-KASH1, or DN-KASH2 were infected with ZIKV at MOI 3. At 24 h p.i., cells and supernatants were collected for western blot or plaque assay. Expression of DN-KASH1 or DN-KASH2 led to reduction of E protein level and viral titers in both control and SUN2$^{KO}$ cells (Fig. 7G and H). These observations indicated that Nesprin-1 and Nesprin-2 are required for actin to interact with viral NS1, thereby for ZIKV replication.

## SUN2 knockout abolishes in vivo ZIKV replication and alleviates neuropathology

To explore whether SUN2 confers a proviral activity in vivo, SUN2 knockout mice (*Sun2$^{-/-}$*) were generated by CRISPR/Cas9 editing technique and genotyped by PCR (Supplementary Fig. 9). Two-day-old WT and *Sun2$^{-/-}$* mice were subcutaneously inoculated with $8×10^5$ PFU of ZIKV or PBS (Fig. 8A). Growth and survival percentages of mock-infected WT and *Sun2$^{-/-}$* mice were comparable, indicating SUN2 depletion does not affect the development of mouse. From 8 days post infection (d p.i.), weights of WT mice gradually declined, while *Sun2$^{-/-}$* mice showed weight gain (Fig. 8B). Survival rate of ZIKV-infected *Sun2$^{-/-}$* mice was significantly elevated (75%) compared to WT mice (0%) at 14 d p.i. (Fig. 8C). As shown in Fig. 8D–G, severe paralysis (Fig. 8D red arrows) and microcephaly (Fig. 8E–G) were observed in the ZIKV-infected WT mice, which was significantly abrogated in *Sun2$^{-/-}$* mice (Fig. 8D–G). At 10 d p.i., neurological symptoms (including reeling gait,

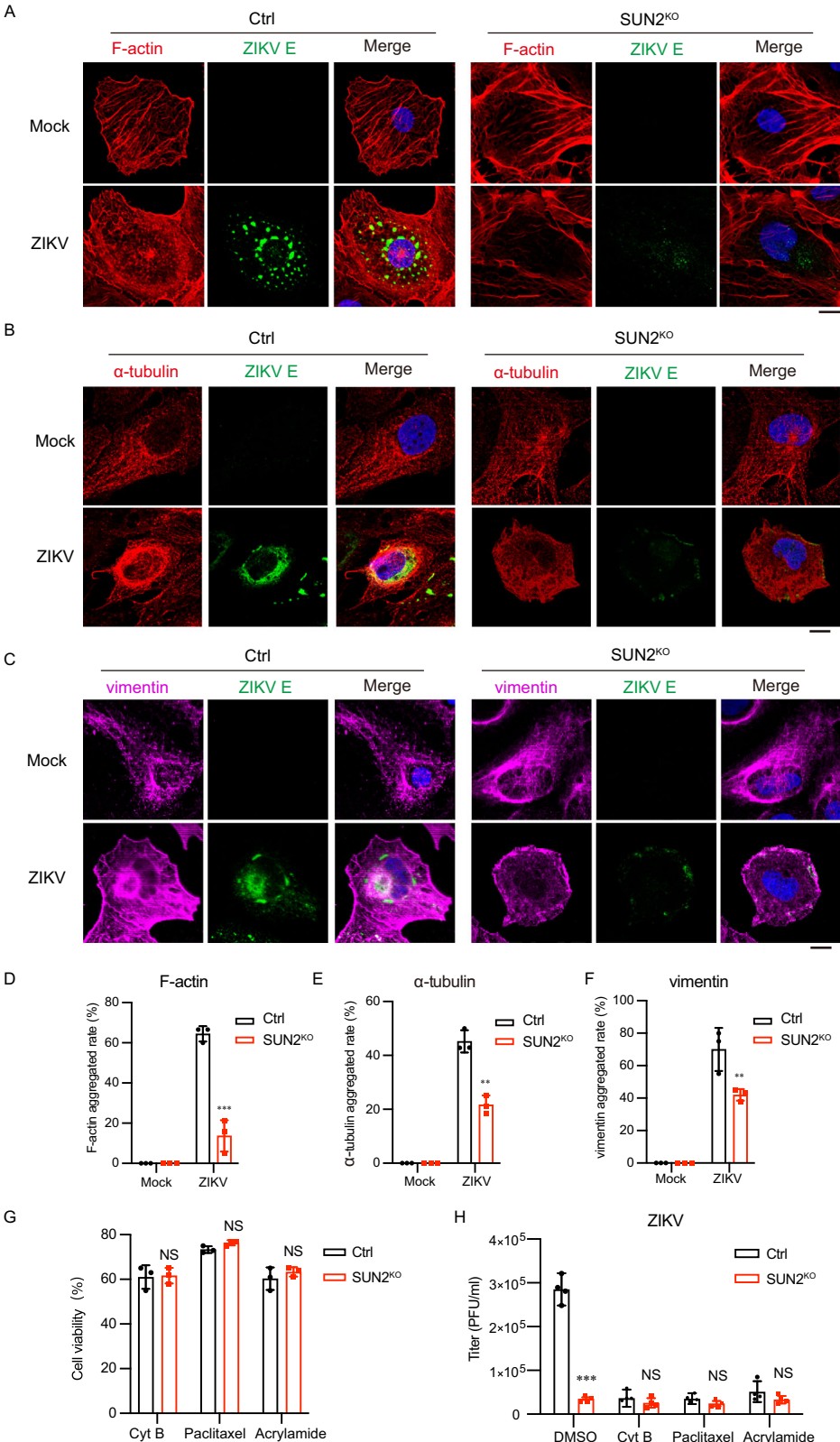

hind-limb paralysis, and dead) of *Sun2*[-/-] mice were dramatically alleviated compared to WT mice (Fig. 8H).

Then, we compared the histological alteration and in vivo replication levels of ZIKV in tissues including brain, lung, liver, and kidney at 10 d p.i. The viral loads in all tissues of *Sun2*[-/-] mice were reduced by 2-3 logs compared to WT mice (Fig. 8I). The H&E staining images showed that destruction of cerebral cortex

integrity of WT mice caused by ZIKV was markedly more severe than *Sun2*[-/-] mice (Fig. 8J). The levels of ZIKV E antigen in cerebral cortex, hippocampus, and striatum of *Sun2*[-/-] mice were dramatically reduced (Fig. 8K). These collective data indicated that in the in vivo neonatal mice, SUN2 plays an essential role in the replication and transmission of ZIKV, as well as the development of neuropathology.

**Fig. 4 | SUN2 regulates the rearrangement of cytoskeleton induced by ZIKV infection. A–C** Images of immunofluorescence labeling for cytoskeletons and ZIKV. Control and SUN2^KO cells were infected with ZIKV at an MOI of 8. At 24 h p.i., cells were stained with anti-ZIKV E antibody (Green) and phalloidin for F-actin (**A**), anti-α-tubulin antibody for microtubule (**B**), or anti-vimentin antibody for intermediate filament (**C**). DAPI stains nucleus (Scale bars, 10 μm). **D–F** Percentages of cells with F-actin, α-tubulin, and vimentin gathered rate in the control and SUN2^KO cells. Results from three independent experiments and at least 10 fields of each view were collected. The data are shown as mean ± SD. Statistical significance was determined using two-tailed unpaired *t* test (**\*\*p < 0.01; \*\*\*p < 0.001*). *p* values: *p* = 0.0005 (**D**), *p* = 0.0017 (**E**), *p* = 0.0024 (**F**). **G** Cytotoxic effect of cytochalasin B,

paclitaxel, or acrylamide was determined by CCK8. The data are shown as mean ± SD, n = 3 biologically independent experiments. Statistical significance was determined using two-tailed unpaired *t* test (NS, not significant, *p* > 0.05). **H** Control and SUN2^KO cells were infected by ZIKV at an MOI of 3, then treated with cytochalasin B (10 μM), paclitaxel (10 μM), or acrylamide (2 μM) from 1 h p.i. respectively. Supernatants were collected at 24 h p.i. for plaque assay. Data are shown as mean ± SD, n = 4 biologically independent experiments. Statistical significance was determined using two-way ANVOA (NS, not significant, *p* > 0.05; ***\*\*\*p < 0.001*). *p* values: (**H**) *p* = 0.0008 (DMSO), *p* = 0.4573 (Cyt B), *p* = 0.2007 (Paclitaxel), *p* = 0.1302 (Acrylamide). Source data are provided as a Source Data file.

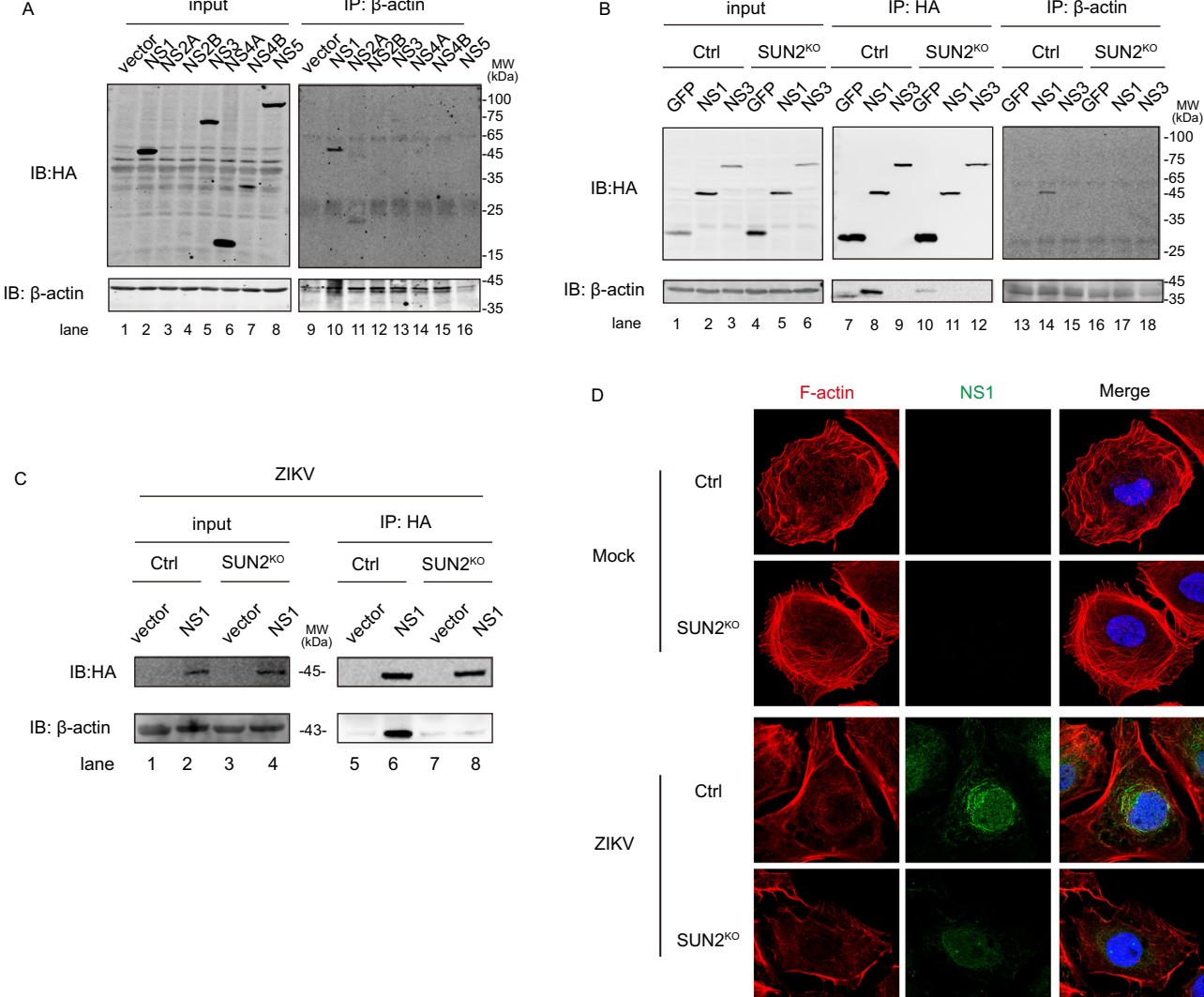

**Fig. 5 | SUN2 is involved in the interaction between ZIKV NS1 and actin. A–C** Co-IP assay. **A** 293 T cells were transfected with plasmids expressing ZIKV HA-NS proteins for 48 h. Whole-cell extracts were prepared for IP assay using anti-β-actin antibody. Samples were detected by western blot using anti-β-actin and anti-HA antibodies. Representative images of three independent experiments are shown. **B** Control and SUN2^KO 293 T cells were transfected with plasmids expressing HA-GFP, HA-NS1, or HA-NS3. The cell lysates were harvested for co-IP assay using anti-HA agarose beads or anti-β-actin antibody. Western blot was performed to detect the interaction. Vector expressing HA-GFP and HA-NS3 were probed as the negative control. Representative images of three independent experiments are shown.

**C** Control and SUN2^KO Huh7 cells were transfected with plasmid expressing HA-NS1, followed by infection with ZIKV at an MOI of 5 at 12 h p.t. Whole-cell extracts were harvested at 24 h p.i. and prepared for IP assay using anti-HA antibody. Samples were detected by western blot using anti-HA and anti-β-actin antibodies. Representative images of three independent experiments are shown. **D** IFM assay. Control and SUN2^KO cells were infected with ZIKV at an MOI of 8. At 24 h p.i., cells were fixed and stained with anti-NS1 antibody (Green) and phalloidin for F-actin (Red). DAPI stains nucleus (Scale bars, 10 μm). Representative images of three independent experiments are shown. Source data are provided as a Source Data file.

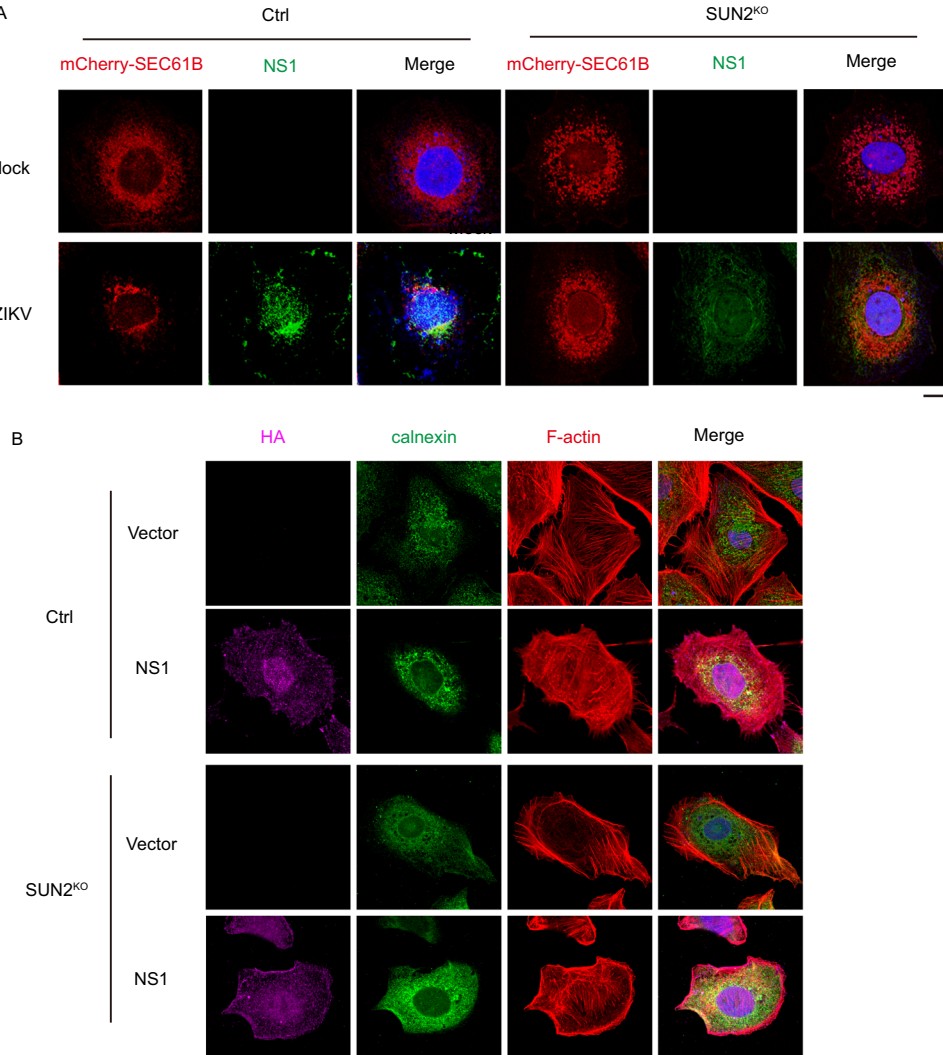

**Fig. 6 | Role of NS1 in the ER and cytoskeleton remodeling induced by ZIKV.
A, B** IFM assay. **A** Control and SUN2[KO] cells were infected with ZIKV at an MOI of 8.
At 24 h p.i., cells were fixed and stained with anti-ZIKV NS1 (Green) antibody. DAPI
stains nucleus (Scale bars, 10 μm). **B** Control and SUN2[KO] cells were transfected with
empty vector or plasmid expressing HA-NS1 for 36 h. Cells were fixed and stained
with anti-HA antibody (Magenta), anti-Calnexin antibody (Green), and phalloidin
for F-actin (Red). DAPI stains nucleus (Scale bars, 10 μm). Representative images of
three independent experiments are shown.

## Discussion

ER membrane remodeling driven by the interaction between flavivirus
proteins and cytoskeleton proteins is critical to the biogenesis of ROs
and viral RNA replication. However, involvement of nuclear proteins in
this step remains unknown. Current study first revealed that nuclear
membrane protein SUN2, coupling with Nesprins, plays a prominent
role in the cytoskeleton rearrangement and RO formation triggered by
ZIKV, through mediating the interaction between ZIKV NS1 and
filament actin.

Interestingly, we found that SUN2 but not SUN1 is a specific host
factor for flaviviruses. In the absence of SUN2, replication of three
tested flaviviruses was inhibited in two human cell lines (ZIKV, DENV,
and JEV) and mouse tissues (ZIKV), indicating that the proviral role of
SUN2 is across the species. As SUN2 is evolutionarily conserved, it will
be interesting to investigate whether SUN2 confers a similar activity
in mosquito cells. On the other hand, SUN1 is dispensable in the
infection of ZIKV and DENV2. Although SUN2 and SUN1 share many
similarities in structures and functions, each has its own unique
functions. For example, SUN1 protein is concentrated at the nuclear
pore complex (NPC) and is important for NPC assembly while SUN2
distribution on the INM is roughly uniform[33], so it is not surprising to

observe that flaviviruses favor co-opting SUN2 rather than SUN1 for
their infection.

Our study underscores an importance of SUN2 integrity in its
proviral function. Without its N-terminal TM region, SUN2 loses its
localization on the nuclear membrane; without CC or SUN domain,
SUN2 cannot form a trimer or connect the ONM Nesprins[21]. None of
three truncates trans-complementation is sufficient to restore the viral
replication, strongly suggesting that SUN2 provides a mechanic sup-
port for flaviviruses. Indeed, our data obtained from the transmission
electron microscopy and IFM assays validated that SUN2 is essential in
the rearrangement of cytoskeletons and formation of ROs induced by
ZIKV. In the ZIKV-infected Huh7 cells, all three classes of cytoplasmic
skeleton network, including actin filaments, microtubules, and inter-
mediate filaments underwent rearrangement, either forming a cage-
like structure or loosely surrounding the nuclei. These observations
were consistent with findings from Bartenschlager's group[16], while a
recent study showed no apparent changes of actin and tubulin in the
ZIKV infected U2OS cells[17]. This discrepancy might be due to different
cell models or virus strains used.

Importantly, our data demonstrated that SUN2 plays a critical role
in the remodeling of cytoskeleton network induced by ZIKV infection,

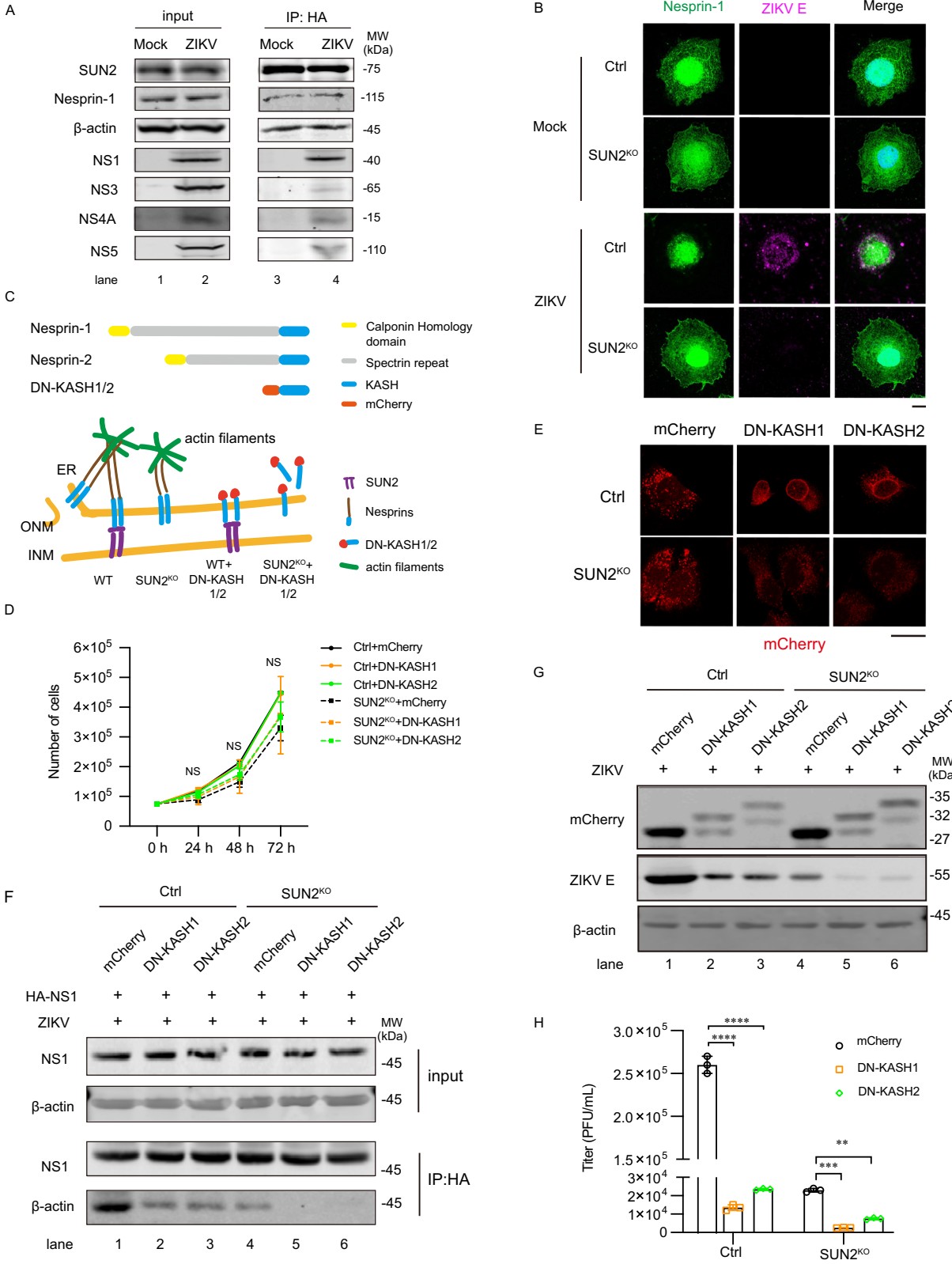

although their loss does not affect the cytoskeletal structure in latent cells. Intriguingly, SUN2 is not involved in the rearrangement of cytoskeleton induced by SFV. During SFV infection, actin filament and microtubule mainly function in the internalization and transportation of ROs from plasma membrane to the perinuclear area[34], thus remodeling of filament and microtubule does not resemble that induced by ZIKV.

The localization of actin and tubulin in the SFV-infected cells is not altered by SUN2 knockout, supporting the observation that SUN2 is not a host factor for SFV. The *trans*-complementation of individual SUN2 truncates is not sufficient to restore the rearrangement of cytoskeleton, probably because movement of cytoskeleton requires a full-length SUN2 for force transmission from nucleus[35]. Our findings that the cytoplasmic

**Fig. 7 | Nesprins mediates the regulation of SUN2 on the NS1-actin association. A** Co-IP assay. The SUN2$^{RES}$ Huh7 cells were infected with ZIKV at an MOI of 5. Whole-cell extracts were harvested at 24 h p.i. and prepared for co-IP assay using anti-HA antibody. Western blot was performed using anti-SUN2, anti-β-actin, anti-Nesprin-1, anti-NS1, anti-NS3, anti-NS4A, and anti-NS5 antibodies. **B** IFM assay. Control and SUN2$^{KO}$ cells were infected with ZIKV at an MOI of 8. At 24 h p.i., cells were stained with anti-ZIKV E (Magenta) and anti-Nesprin-1 (Green) antibodies. DAPI stains nucleus (Scale bars, 10 μm). Representative images of three independent experiments are shown. **C** Upper panel, schematic representation of Nesprin-1, Nesprin-2, and DN-KASH1/2. Lower panel, association between SUN2, Nesprins, and actin in the control and SUN2$^{KO}$ expressing DN-KASH1 or 2. **D** Cell growth curve. Control and SUN2$^{KO}$ cells expressing mCherry, DN-KASH1, and DN-KASH2 were counted at 0, 24, 48, and 72 h post inoculated. The data are shown as mean ± SD, n = 3 biologically independent samples. **E** IFM assay. Control and SUN2$^{KO}$ cells expressing mCherry, DN-KASH1, and DN-KASH2 cells were infected with ZIKV at an MOI of 8. At 24 h p.i., DAPI stains nucleus (Scale bars, 10 μm). Representative images of three independent experiments are shown. **F** Co-IP assay. Control and SUN2$^{KO}$ cells expressing mCherry, DN-KASH1, and DN-KASH2 were transfected with plasmid expressing HA-NS1, followed by infection with ZIKV at an MOI of 5. Whole-cell extracts were harvested at 24 h p.i. for co-IP assay using anti-HA antibody. Samples were detected by western blot using anti-NS1 and anti-β-actin antibodies. **G, H** ZIKV replication levels. Control and SUN2$^{KO}$ cells expressing mCherry, DN-KASH1, and DN-KASH2 were infected with ZIKV at an MOI of 3. At 24 h p.i., cells and supernatants were harvested for western blot (**G**) and plaque assay (**H**). **H**, Data are shown as mean ± SD, n = 3 biologically independent experiments. Statistical significance was determined using two-way ANVOA (**$p < 0.01$; ***$p < 0.001$; ****$p < 0.0001$). $p$ values: (**H**) WT cells: $p < 0.0001$ (DN-KASH1), $p < 0.0001$ (DN-KASH2); SUN2$^{KO}$ cells: $p = 0.0002$ (DN-KASH1), $p = 0.0019$ (DN-KASH2). Source data are provided as a Source Data file.

part of Nesprin-1 moves to the perinuclear region and colocalizes with viral protein upon ZIKV infection in a SUN2-dependent manner (Fig. 7B) demonstrated that the virus-induced cytoskeleton movement needs a mechanical support from LINC proteins.

Furthermore, we detected a previously unknown interaction between ZIKV NS1 and actin, as well as ZIKV E and vimentin. Experimentally, our co-IP data indicated an interaction between NS1/actin (Fig. 5B–D), and AlphaFold2 analysis predicted three potential interaction interfaces (Supplementary Fig. 10). Nonetheless, as NS1 mainly resides in the ER lumen while actin is cytoplasmic, other proteins might be involved in their interaction. Whether NS1 directly or indirectly binds to actin needs to be further investigated. To date, many associations between flavivirus and cytoskeleton proteins have been reported, including DENV NS4A and vimentin[19], WNV NS1 and actin[15], ZIKV E and actin[11], and JEV NS3 and actin[33]. Given viral nonstructural proteins including NS4A, NS4B, and NS1 are closely associated with each other[36], and cytoskeleton proteins are highly interconnected[37], it might be not critical for certain cytoskeleton protein to be recruited by certain viral protein. Virtually, the SUN2 depletion disrupts not only remodeling of actin filaments, but also remodeling of microtubules and intermediate filaments despite it does not directly modulate the interaction of E/vimentin. Therefore, combined with the observations that both SUN2 depletion and Nesprins DN expression impair the interaction between NS1 and actin, we propose that SUN2 is the major INM protein to direct the association of NS1/actin through ONM Nesprins, while SUN1 might be simultaneously involved in the association of E/vimentin. To be noted, expression of either DN KASH1 or KASH2 in the control cells alone is sufficient to disrupt the interaction of NS1/actin and viral replication (Fig. 7E–G), implying that Nesprin-1 and Nesprin-2 could not be substituted by each other. Interestingly, the DN KASH1 and KASH 2 expression on top of SUN2 depletion, almost abolish the NS1-actin interaction and viral replication, illustrating a critical role of LINC in supporting viral replication.

Our study revealed that a biological significance of NS1-actin association is to assist in driving remodeling of cytoskeleton. Although NS4A and NS4B have been implicated to induce ER membrane bending, they are not sufficient to form ROs resembling the structures induced by virus infection[4,5], suggesting other viral proteins such as NS1 are also required. NS1 is originally located in the ER lumen but soon redistributed to three destinations including viral replication sites[31] to promote viral RNA synthesis[6,31]. We found that NS1 is colocalized with ER membrane (Fig. 6A), and expression of NS1 alone was sufficient to lead ER membrane curvature and build ROs in vitro (Fig. 6B), illustrating one of mechanisms by which NS1 facilitates viral RNA synthesis is through physically recruiting actin, and hence driving rearrangement of cytoskeleton and formation of ROs.

In summary, we proposed that SUN2, together with Nesprins, promote the flaviviral RNA synthesis, via regulating the NS1-actin mediated cytoskeleton remodeling and formation of ROs (Fig. 9).

Apparently, the mechanism that SUN2 facilitates flavivirus replication differs from the mechanisms of its role in nucleic-replicating viruses, by which it mainly acts as a "doorkeeper" to regulate the nuclear import or egress of HIV and HSV[24,25]. These findings provide important insights into the underlying mechanism of LINC proteins including SUN2 and Nesprins in the infection of flaviviruses, unveiling a potential anti-viral target for the design of therapeutics drugs.

## Methods

### Ethics statement

The use of laboratory animals and animal-related experiments was reviewed and approved by the Sun Yat-sen University Institutional Animal Care and Use Committee (SYSU IACUC) (approval number is 2021-000473). All animal-related experiments were performed in accordance with animal research reporting of in vivo experiments guidelines approved by the SYSU IACUC and were conducted in the laboratory designed to ensure biological safety.

### Cell culture and reagents

Human hepatoma cells (Huh7) cells were provided by Dr. Yi-Ping Li (Sun Yat-sen University). Huh7 cells, Human lung carcinoma epithelial cells [A549, American Type Culture Collection (ATCC), CCL-185], human cervical cancer cells (HeLa, ATCC, CCL-2), and human embryonic kidney cells (293 T, ATCC, CRL-3216) were maintained in Dulbecco's modified Eagle medium (DMEM, Sigma) supplemented with 10% fetal bovine serum (FBS, Sigma). African green monkey kidney cells (Vero, ATCC, CCL-81) and baby hamster kidney cells (BHK-21, ATCC, CCL-10) were maintained in DMEM supplemented with 5% FBS. The media were added with 100 units/ml of streptomycin and penicillin (Invitrogen). All cells were cultured at 37 °C in 5% $CO_2$. The *Aedes albopictus* cells C6/36 (ATCC, CRL-1660) were cultured in RPMI 1640 (Thermo Fisher Scientific, 11875093) medium containing 10% FBS at 28 °C. Puromycin, Lipofectamine® 2000, and blasticidin were purchased from Sigma-Aldrich, Invitrogen, and InvivoGen respectively.

### Viruses

ZIKV strain H/PF/2013 (GenBank: KJ776791), DENV2 (DENV2 16681), and JEV (14-14-2 vaccine strain) were obtained from Guangzhou Centers for Disease Control and propagated in Vero cells or C6/36 cells respectively. EMCV and SFV were kindly provided by Prof. Rongbin Zhou (University of Science and Technology of China) and Prof. Xi Zhou (Wuhan Institute of Virology), and propagated in HeLa and C6/36 cells respectively. VSV was propagated in Vero cells. All viral stocks were clarified by centrifugation, aliquoted, and stored at −80 °C until use.

### Virus infection

Cells were infected with viruses at an MOI of 3 to ensure that most cells were infected unless specifically indicated. For ZIKV, DENV2, JEV, SFV,

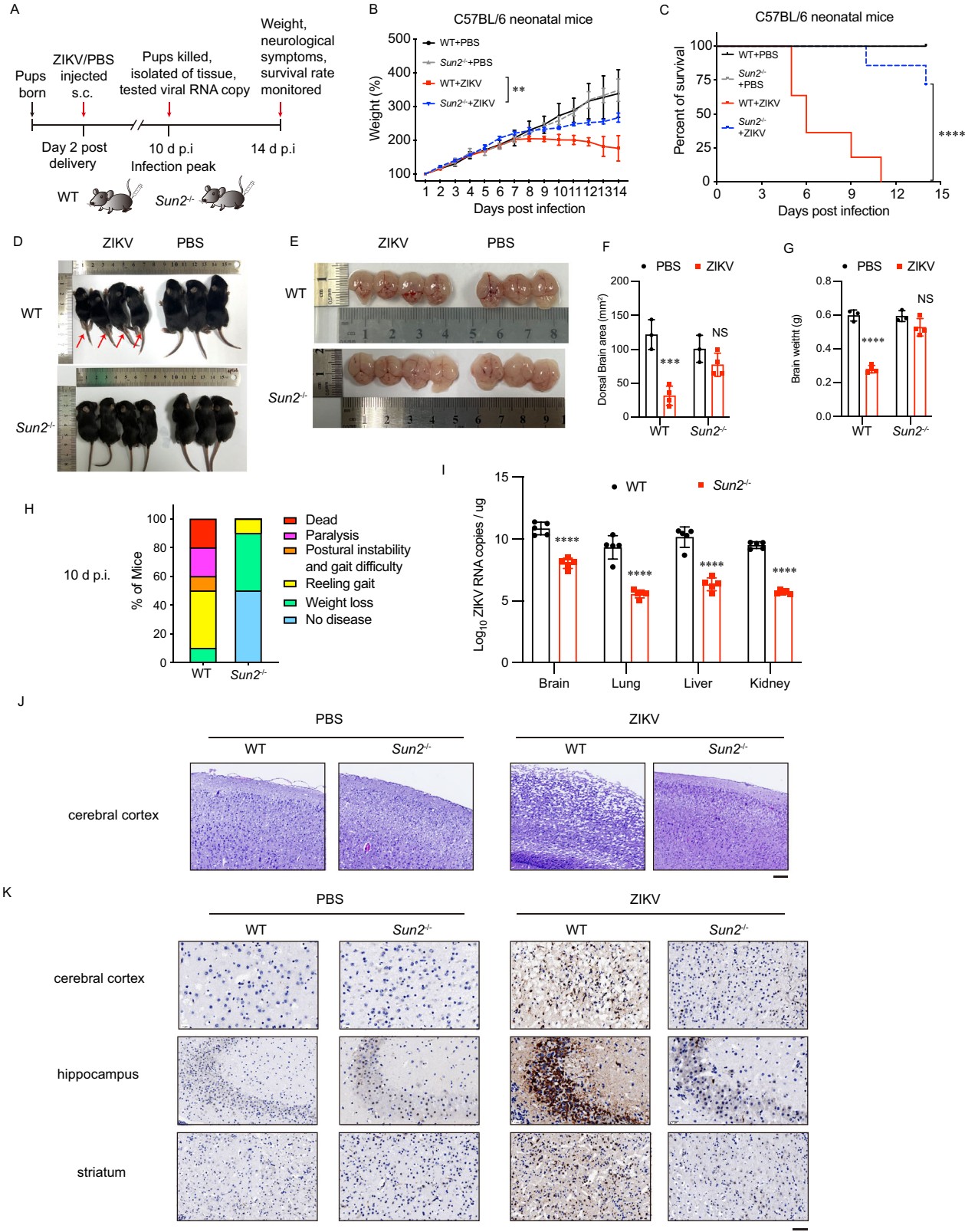

**Virus titration**

Virus titers were determined by standard plaque assay. Serial 10-fold dilutions of each sample were prepared, and 100 μL/well of the diluted virus were applied. The cells were cultured at 37 °C for 1 h and intermittently rocked every 15 min to allow the adsorption of virions. The incubation medium was removed and cultured in the mixture of 2×MEM (Invitrogen) and isopycnic 2% low melting point agarose

and VSV, supernatants were harvested depending on the experiment endpoint, and cell debris was removed by centrifugation at 185 x g for 5 min and filtration. For EMCV, both supernatants and cells were collected, followed by three cycles of freezing-and-thawing and centrifugation at 185 x g for 5 min. In the multi-step growth assay, cells were infected with ZIKV at an MOI of 0.01. At 24, 48, and 72 h p.i., supernatants were collected for virus titration.

**Fig. 8 | SUN2 knockout impairs the ZIKV replication and neuropathology of neonatal mice. A** Experimental scheme. Two-day-old WT and *Sun2⁻/⁻* C57BL/6 neonatal mice were subcutaneously inoculated with 50 μL PBS or ZIKV suspension (8 ×10⁵ PFU). **B** Weight growth. C57BL/6 neonatal mice were weighed daily over 14 days after ZIKV injection. Weights were expressed as percentage of body weight. Results shown are the mean ± SD. Weight changes were compared using two-way ANOVA. **\*\*P < 0.01, p = 0.0051, n = 10 mice/group. **C** Survival curves. Lethality was monitored for 14 days. Kaplan-Meier survival curves were analyzed by the log-rank test. \*\*\*\*P < 0.0001, n = 10 mice/group. **D** Representative images of WT and *Sun2⁻/⁻* mice and brains captured at 10 d p.i. **E** Brain sizes of WT and *Sun2⁻/⁻* mice captured at 10 d p.i. **F** Quantification of dorsal brain area was shown in the histogram. Bars represent means ± SD compared using two-way ANOVA (NS, no statistical significance; \*\*\*P < 0.001. *p* values: *p* = 0.0001 (WT); *p* = 0.2978 (*Sun2⁻/⁻*). N = 3 for PBS, or n = 4 for ZIKV mice/group. **G** Quantification of brain weights of WT and *Sun2⁻/⁻* mice at 10 d p.i. Bars represent means ± SD compared using two-way ANOVA (NS, no statistical significance; \*\*\*\*P < 0.0001. *p* values: *p* < 0.0001, (WT); *p* = 0.0791,

(*Sun2⁻/⁻*). N = 3 for PBS, or n = 4 for ZIKV mice/group). **H** Neurologic symptoms. Signs were assessed at 10 d p.i. The percentages of WT and *Sun2⁻/⁻* mice displaying indicated signs are shown. **I** ZIKV RNA copies in mice tissues. At 10 d p.i., tissues including brain, lung, liver, and kidney were collected, homogenized, and analyzed by qRT-PCR. Viral RNA copies were shown as viral RNA equivalents per μg upon comparing with a standard curve produced by gradient 10-fold dilutions of ZIKV RNA. Results were from three independent experiments with a total of 5 mice per group. Bars represent means ± SD compared using two-way ANOVA; \*\*\*\*p < 0.0001 (Brain, lung, liver, kidney). **J** H&E staining of mouse brain cerebral cortex (scale bar, 10 μm). Representative images of three independent experiments are shown. **K** Immunohistochemistry staining of mouse brain tissues. Representative ZIKV immunolabeling (anti-ZIKV E antibody) in the cerebral cortex, hippocampus, and striatum regions from mice at 10 d p.i. Scale bar, 20 μm. Representative images of three independent experiments are shown. Source data are provided as a Source Data file.

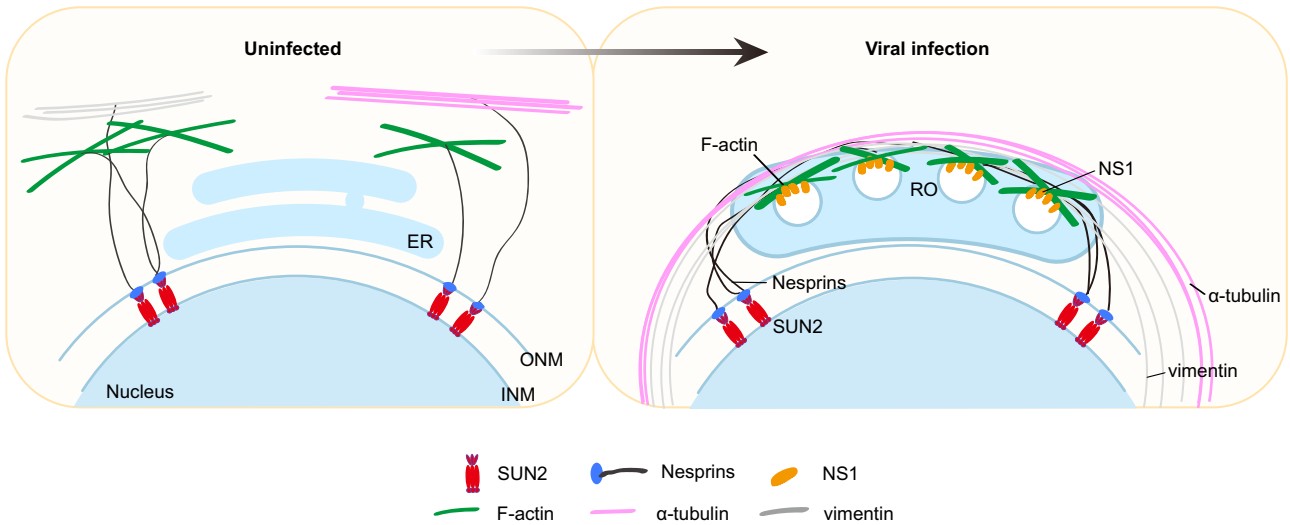

**Fig. 9 | A proposed model to illustrate the mechanism by which SUN2 promotes flavivirus infection.** Nuclear membrane protein SUN2 coupling with Nesprins connect to the cytoskeleton including filament, microtubule, and intermediate filament. Upon flaviviral infection, SUN2 directs the movement of filament driven by viral NS1-actin association through Nesprins, hence facilitating the cytoskeleton remodeling and formation of ROs. Therefore, SUN2 promotes the viral RNA synthesis and infection.

(Sangon Biotech) (1:1) or 2% methylcellulose (1:1) (Sigma-Aldrich). Visible plaques were counted at 2 to 3 days (SFV, VSV, EMCV), 4-5 days (ZIKV, JEV), or 5-7 days (DENV2). Cells were fixed with 10% formaldehyde and plaques were visualized by staining with 1% crystal violet.

## Antibodies

Primary antibodies used in western blot, immunofluorescence, and immunohistochemistry (IHC) were listed in Supplementary Table 1. Secondary antibodies include IRDye 800 CW-conjugated anti-rabbit IgG, IRDye 680 CW-conjugated anti-mouse IgG (LI-COR), and horseradish peroxidase (HRP)-conjugated anti-mouse IgG (CST). Secondary antibodies used in immunofluorescence assay included goat anti-rabbit IgG secondary antibody (Alexa Fluor 488) and goat anti-mouse IgG secondary antibody (Alexa Fluor 647) from Invitrogen.

## Western blot

Cells were harvested and lysed with RIPA lysis buffer [pH 7.4, 50 mM Tris-HCl, 0.5% (vol/vol) NP-40, 1% Triton X-100, 150 mM NaCl, 1 mM EDTA, 1 mM phenylmethylsulfonyl fluoride (PMSF), 1% protease inhibitor cocktails, 1 mM $Na_3VO_4$, and 1 mM NaF]. Samples were separated on SDS-PAGE and transferred onto nitrocellulose membranes. Membranes were blocked in 5% BSA in 0.1% TBST. Primary antibodies were diluted in 0.1%TBST with 3% BSA and incubated overnight at 4 °C.

Detection was performed with secondary antibodies. Immunoreactive bands were visualized using an Odyssey IR imaging system (LI-COR) as described previously[38]. The Quantity One program (Bio-Rad) was used to quantify the western blot results.

## qRT-PCR

Reverse transcription was performed using HI Script Q RT SuperMix (Vazyme). qRT-PCR was performed using a Bio-Rad CFX96 machine. Primers used for qRT-PCR were listed in Supplementary Table 2. qRT-PCR data were analyzed by calculating ΔΔCt values as described previously[39].

## Plasmid construction

Oligonucleotide sequences of single guide RNAs targeting *SUN2* or *SUN1* were listed in Supplementary Table 3. Oligonucleotides were annealed and inserted into plasmid vector lentiCRISPR v2 (Addgene, #52961). Positive plasmids were designated as pLenti-sgSUN2#1, pLenti-sgSUN2#2, pLenti-sgSUN1#1, and pLenti-sgSUN1#2.

Fragments of full-length of SUN2, rescue mutant, or truncates with deleted N-terminal and transmembrane domain (ΔN + TM), coiled helix domain (ΔCC), or SUN domain (ΔSUN) were amplified by PCR using SUN2 cDNA template (kindly provided by Prof. Jiahuai Han at Xiamen University). Primer sequences were listed in Supplementary Table 4. PCR fragments were purified and cloned into pSG5 or lentiviral

vector pLV-EF1α-IRES-blasticidin (Addgene, #85133). Positive clones were verified by DNA sequencing.

Plasmids expressing ZIKV E protein and NS proteins (NS1, NS2A, NS2B, NS3, NS4A, NS4B, NS5) were kindly provided by Prof. Guigen Zhang at Sun Yat-sen University and used for PCR templates. Viral genes (*E, NS1, NS2A, NS2B, NS3, NS4A, NS4B, NS5*) (SZ01, GenBank: KU866423) were amplified by PCR and inserted into the *Eco*R I (Sac I for NS5) and *Xho* I sites of pSG5 plasmid with HA fused with their N-termini. Sequences of primers were listed in Supplementary Table 4.

Gene fragments (*SEC61B*, KASH domains of *Nesprin-1* and *Nesprin-2*) were amplified by PCR using A549 cDNA as template. mCherry was amplified from pLVX-mCherry-N1 (YouBio, VT2003) by PCR. Primer sequences were listed in Supplementary Table 4. PCR fragments were purified and cloned into pLV-EF1α-IRES-blasticidin with mCherry fused with its N terminus.

### Generation of knockout cells by CRISPR/Cas9 gene editing
293 T cells were transfected with sgRNA vector (pLenti-sgSUN2#1, pLenti-sgSUN2#2, pLenti-sgSUN1#1, or pLenti-sgSUN1#2.) and two packaging plasmids pSPAX2 (Addgene, #12260) and pMD2.G (Addgene, #12259) using FuGENE HD transfection reagent (Promega). Supernatants were collected at 2 days post-transfection and passed through a 0.45-µm filter. Subsequently, lentivirus supernatants were transduced into Huh7 cells, A549 cells, or 293 T cells. 24 hours later, cells were transferred to 10-cm dishes and selected with 1 µg/ml of puromycin for 10-14 days to isolate single-cell clones. Puromycin-resistant clones were sorted and confirmed by western blot and DNA sequencing. Genomic DNA was extracted using a cell genomic DNA extraction kit (Bioteke). Regions surrounding sgRNA target sequences were amplified by PCR. PCR products were then cloned into pMD-18T (TaKaRa) for DNA sequencing as described previously[38].

### Generation of SUN2[RES], SUN2[ΔN+TM], SUN2[ΔCC], SUN2[ΔSUN], mCherry-SEC61B, DNKASH1 or DN-KASH2-expressing cells
293 T cells were transfected with plasmids (pLV-res-sgSUN2#1, pLV-SUN2-ΔN + TM, pLV-SUN2-ΔCC, pLV-SUN2-ΔSUN, pLV-mCherry-SEC61B, pLV-mCherry-Nesprin-1 KASH, or pLV-mCherry-Nesprin-2 KASH) and two packaging plasmids pSPAX2 and pMD2.G using FuGENE HD transfection reagent (Promega). Supernatants were collected at 2 days post-transfection and passed through a 0.45-µm filter. Subsequently, SUN2[KO] cells were transduced with lentiviruses carrying full length of SUN2, ΔN + TM, ΔCC, or ΔSUN. Or, control and SUN2[KO] cells were transduced with lentiviruses carrying mCherry-SEC61B, DN-KASH1 or DN-KASH2. Stable cell populations were generated by selection and expansion in the presence of 15 µg/ml blasticidin for 7 to 10 days.

### Immunofluorescence (IFM) and confocal microscopy
Control, SUN2[KO], SUN2[RES], SUN2[ΔN+TM], SUN2[ΔCC], and SUN2[ΔSUN] cells were infected with ZIKV at an MOI of 8, and fixed at 24 h p.i.. Control and SUN2[KO] cells were infected with SFV at an MOI of 3 and fixed at indicated times. Briefly, cells were fixed with 4% PFA after 3 washes with PBS, and permeabilized with 0.02% Triton-X 100 for 15 minutes. Then, cells were blocked with 5% BSA for 30 minutes, followed by incubation with primary antibody at 4 °C overnight. After washes, cells were incubated with Alexa Fluor 647-conjugated goat anti-mouse-IgG (Invitrogen) or Alexa Fluor 488-conjugated anti-rabbit-IgG (Invitrogen) in PBS at room temperature (RT) for 1 h. Cells were then stained with TRITC Phalloidin (Yeasen) diluted in PBS for 45 minutes, followed by incubation with DAPI (Invitrogen) diluted in PBS for 20 minutes. Fluorescence images were acquired with a Nikon C2 microscope and analyzed with the NIS Elements software.

### Virus entry assay
In virus binding assay, cells were pre-chilled on ice for 10 min before ZIKV infection. Cells were washed two times with ice-cold PBS and

incubated with ZIKV at an MOI of 3 with 2% FBS in ice-cold DMEM, followed by incubation on ice for 45 min for virion binding. After five washes of PBS, cells were lysed to harvest total RNA in TRIzol reagent (Thermo Fisher #15596018). In virus internalization assay, cells were incubated with ZIKV at an MOI of 3 at 37 °C for 45 min. Media were removed, and cells were treated with 400 µg/ml protease K after 3 washes of PBS and kept on ice for 45 minutes. After three washes of PBS, cells were lysed in TRIzol reagent for RNA extraction. Reverse transcriptase PCR (RT-qPCR) was conducted to quantify viral RNA and internal control *β-actin*.

### Replicon assay
The ZIKV replicon plasmids pFK-SGR and pFK-SGR-GDD were kindly provided by Prof. Gang Long at Institution Pasteur of Shanghai Chinese Academy of Science[40]. Briefly, the pFK-SGR and pFK-SGR-GDD were linearized with Mlu I (New England Biolabs), and extracted by phenol-chloroform and precipitated with sodium acetate. The linearized DNA template was transcribed using mMESSAGEmMACHINE® T7 kit (Thermo Fisher Scientific, AM1344) according to the manufacturer's protocol. The product of transcription was purified by lithium chloride (LiCl) precipitation. Replicon RNAs (0.2 µg/well) were transfected into cells using Lipofectamine® 2000 reagent (Invitrogen). Cells were washed with PBS and lysed using passive lysis buffer (Promega) at indicated time points. Luciferase activities were measured using a GLOMAX™ 96 microplate luminometer (Promega).

The pBAC-SARS-CoV-2-replicon-Luciferase plasmid (provided by Prof. Rong Zhang at Fudan University) was transcribed as described previously[41]. Briefly, SARS-CoV-2 replicon RNA was transcribed by the mMESSAGE mMACHINE T7 transcription kit (Thermo Fisher Scientific, AM1344) according to the manufacturer's protocol. The DNA templates were removed by adding Turbo DNase and incubating the mixture at 37 °C for 15 min. The RNA was extracted by LiCl precipitation. The replicon RNA was stored at −80 °C. The replicon RNAs (20 ng/well) were transfected into cells in a 96-well plate using 0.12 µL Lipofectamine® 2000 reagent following the manufacturer's instructions. At 24 h p.i., supernatants were collected and added in the mixture of Nano-Glo® Luciferase buffer and substrate (N1120, Promega) (100:1) at RT for 15 minutes. All the liquid was transferred to an opaque plate and chemiluminescence value was detected using a GLOMAX™ 96 microplate luminometer (Promega).

### Transmission electron microscopy (TEM)
Cells were fixed with 2.5% glutaraldehyde for 1 hour at RT. Samples were incubated with 1% osmium tetroxide for 1 hour at RT and washed with PBS for three times. Samples were dehydrated in a graded ethanol series (from 30% to 100%) at RT and washed with 0.1 M PBS for three times. Then, samples were embedded in epoxy resin with acetone (v/v, 1:1) for 60 minutes at RT and polymerized for at least 48 hours at 60 °C. Ultrathin sections of 70 nm were obtained by Leica EM UC7 ultra-microtome (Leica Microsystems) and a diamond knife. The sections were collected on naked grids and counterstained using lead citrate and uranyl acetate. The grids were examined with a JOEL JEM-1400Flash electron microscope and the images were recorded with a CCD camera (Morada G3, Emsis).

### Treatment of inhibitors
Cytochalasin B (MCE), paclitaxel (MCE), and acrylamide (Sigma) were dissolved in DMSO at a stock concentration of 10 mM. Cells were infected with ZIKV at an MOI of 3, and then incubated with media containing 0.1% DMSO or individual inhibitors (10 µM cytochalasin B, 10 µM paclitaxel or 2 µM acrylamide).

### Co-IP assay
In Fig. 5A and Supplementary Fig. 8A, 293 T cells were transfected with empty vector (pSG5) or plasmids expressing HA-NS1, HA-NS2A, HA-

NS2B, HA-NS3, HA-NS4A, HA-NS4B, and HA-NS5 by Lipofectamine® 2000 reagent. In Fig. 5B and Supplementary Fig. 8C, 293 T cells were transfected with plasmids expressing ZIKV HA-NS1, HA-NS3, or HA-E. In Fig. 5C, Fig. 7F, and Supplementary Fig. 8D, Huh7 cells were transfected with plasmid expressing ZIKV HA-NS1, or HA-NS4A, and at 12 h p.t., cells were infected with ZIKV at an MOI of 5. At 36 h post-transfection or 24 h post ZIKV infection, cells were resuspended in ice-cold RIPA lysis buffer containing protease inhibitors (Sigma) and phosphatase inhibitors (NaF and $Na_3VO_4$). Immunoprecipitation was performed with anti-HA beads (Sigma-Aldrich) or with various antibodies and protein A/G-agarose (Millipore) at 4 °C overnight. Beads were washed by six times, and precipitated proteins were eluted by boiling in loading buffer.

## Cell viability assay
The cell viabilities were analyzed with Cell Counting Kit-8 (CCK8) (HY-K0301, MCE) according to the manufacturer's instruction. Reagent (10 μL) was added to cell culture (96-well plate) and incubated at 37 °C. One hour later, the OD value was measured at 450 nm using a BioTek instrument.

## Flow cytometry analysis
Control and $SUN2^{KO}$ A549 cells were infected with ZIKV at an MOI of 3. At 24 h p.i., cells were suspended in PBS and incubated with ZIKV E antibody (1:500, #GTX634155, GeneTex), followed by incubation with goat anti-mouse IgG secondary antibody (Alexa Fluor 647) (Invitrogen). Then, labeled cells were examined by flow cytometry (Beckman Coulter, CytoFLEX S).

## Animal models and experiments
Wild type and $Sun2^{-/-}$ C57B/L6 mice were purchased from Gem-Pharmatech Co., Ltd and maintained under specific-pathogen-free conditions at the research animal facility of Sun Yat-sen University. Neonatal C57B/L6 mice were breastfed and divided into PBS- and ZIKV-infected group. Two-day-old neonatal WT and $Sun2^{-/-}$ C57B/L6 mice were inoculated with 50 μL PBS or ZIKV ($8x10^5$ PFUs) by subcutaneous injection (s.c.), and monitored for 14 days. All animals were cage-bred with the mouse mothers during the experiment and observed daily until the end of assay. All mice were euthanized by cervical dislocation after the appearance of severe clinical signs, including lethargy and paralysis.

Tissues including brain, lung, liver, and kidney of mice were collected at 10 d p.i. Whole tissue homogenates were prepared using an automated homogenizer. Tissues were ground with PBS buffer and centrifuged at 12,000 × g for 15 min at 4 °C. The supernatants were collected and stored at −80 °C. Total RNA was extracted using TRIzol reagent (Invitrogen) according to the manufacturer's instructions. ZIKV RNA copy analysis was performed using GraphPad Prism 9.0 software.

## Genotyping of Sun2 knockout mice
Mouse tails were collected for DNA extraction using Tissue genome DNA extraction kit (BioTeke, DP1902) and used as template for PCR. Sequences of two pairs of primers were listed in Supplementary Table 5. The PCR products were detected by DNA gel.

## Haemotoxylin and Eosin (H&E) and immunohistochemistry (IHC) staining
Mice brains were collected at 10 d p.i. and immediately fixed in 4% PFA. After a serial alcohol gradient, brain sections were embedded in paraffin wax blocks. For H&E staining, tissue was dewaxed in xylene, rehydrated through decreasing concentrations of ethanol (from 100% to 75%) and washed in PBS. Tissue samples were stained by H&E using standard procedures, and dehydrated through increasing concentrations of ethanol and xylene.

For IHC staining, mice were transcranially perfused with cold PBS followed by ice-cold 4% PFA. Brains were removed and post-fixed for 24 h, and embedded in paraffin after dehydration and diaphanization. Brains were cut with a microtome into a thickness of 3-5 mm slice, followed by deparaffinization and washing with PBS. Antigens were restored by boiling in 1xImproved Citrate Antigen Retrieval Solution (Beyotime Biotechnology) for 20 min. After washing with PBS for three times, endogenous peroxidase inhibitor and non-specific stain inhibitor (UltraSensitiveTM SP IHC Kit, MXB, KIT-9710) were added to the specific area circled by IHC pen, and incubated at RT for 10 min respectively. Slides were blocked in 5% BSA in PBS and permeabilized with 0.2% Triton X-100, followed by incubation with ZIKV envelope rabbit polyclonal antibody (1:2000, GeneTex) overnight at 4 °C. After washing with PBS, slides were incubated with streptavidin-biotin-peroxidase (Ultra-SensitiveTM SP IHC Kit, MXB, KIT-9710) for 15 min. Slides were then covered with 1×DAB (MXB, DAB-0031/1031) for 20-30 sec. Reactions were stopped by distilled water. Counter-staining was performed with hematoxylin staining solution. Automatic digital slide scanning system (ZEISS) AxioScan.Z1 was used for image acquisition and analysis.

## Statistical analysis
Data were analyzed with GraphPad Prism 9.0 software. All the statistical analyses were using one-way ANOVA, two-way ANOVA, or an unpaired, two-tailed Student's *t*-test. All the data were shown as means ± standard deviations (SD), the data points and bar graphs represented the mean of independent biological replicates. Kaplan-Meier survival curves were analyzed by the log-rank test, and weight changes were compared using two-way ANOVA.

## Reporting summary
Further information on research design is available in the Nature Portfolio Reporting Summary linked to this article.

## Data availability
All data are included in the article and Supplementary figures. Source data are provided with this paper.

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

## Acknowledgements

Grants from the National Natural Science Foundation of China (82271385 to P.Z., 31970887 to P.Z.), Guangzhou Municipal Science and Technology Program (202206010114 to P.Z., 2023A04J2235 to C.C.), Natural Science Foundation of Guangdong Province (2021A1515011491 to P.Z., 2022A1515010451 to C.L.), Guangdong Basic and Applied Basic Research Foundation (2023A1515010476 to C.C.). We thank Prof. Qinfen Zhang at Sun Yat-sen University for her assistance of transmission electron microscopy. The funders had no role in study design, data collection and analysis, the decision to publish, or preparation of the manuscript.

## Author contributions

Y.H., Q.P., X.T., C.C., X.Z., C.H., and Z.H. performed the experiments. Y.H., Q.P., Y.L., C.Y., C.L., and P.Z. designed the experiments. Y.L., C.Y., C.L., and P.Z. provided administrative, supervision, technical, or material support. Y.H., Q.P., X.T., Y.L., C.Y., C.L., and P.Z. performed data analysis. Y.H., C.L., and P.Z. wrote the draft of the manuscript, with the other authors contributing to editing into the final form.

## Competing interests

The authors declare no competing interests.
