## [Peer Review File · Nature Communications]

REVIEWER COMMENTS

Reviewer #1 (Remarks to the Author):

In this manuscript, Huang et al. reported that the nuclear envelope protein, SUN2, plays a critical role in viral genome replication through its involvement in modulating the cytoskeleton and facilitating the formation of viral replication organelles. Initially, the authors demonstrated that cells lacking SUN2 exhibited a decrease in viral growth, which can be attributed to the impairment of viral replication organelle formation. Next, they showed that the F-actin fiber underwent re-organization around the viral replication site, which was found to be absent in cells lacking SUN2. Then, the authors demonstrated that NS1 interacts with beta-actin in a SUN2-dependent manner. Finally, in vivo experiments were conducted using a mouse model, wherein it was observed that the virulence of ZIKV was diminished in cells lacking SUN2. Overall, they clearly show the involvement of SUN2 in ZIKV genome replication. However, despite its significance, there is a dearth of research on the molecular mechanisms underlying SUN2's involvement in these processes. It appears that addressing this critical point is essential for the acceptance of the manuscript in Nature Communications.

Figure 1. To ensure the proper propagation of the virus, the author must demonstrate the impact of SUN2 knockout on cell growth, as the growth retardation of host cells can significantly impede viral propagation.

Figure 2. The imaging quality is inadequate, making it imperative to investigate the alterations in SUN2 localization following viral infection. Moreover, it is crucial to demonstrate changes in nesprin localization as well. Without such evidence, it would be difficult to discuss the role of SUN2 at the nuclear envelope in the formation of the viral replication organelle. To propose a mechanistic model for SUN2/cytoskeleton-mediated replication organelle formation, the author should conduct experiments using deletion mutants of SUN2.

Figure 3. To assess alterations in the morphology of ER-derived replication organelles, it is imperative to provide fluorescent images depicting changes in the shape of the ER membrane upon virus infection, utilizing cells expressing fluorescent protein-fused ER markers.

Figure 4. A quantitative analysis is necessary to assess changes in cytoskeletal organization. They only presented single-cell images. Although the results presented in Fig.4E depict the entire viral life cycle, they do not specifically demonstrate viral genome replication. Hence, the author should conduct identical experiments using a replicon system.

Figure 5. The experiments conducted do not reveal any interactions between the virus and SUN2. Therefore, to elucidate the molecular mechanism underlying SUN2's role in viral replication organelle formation, the authors must show the evidence indicating a physiological link between SUN2 and viral components.

Reviewer #2 (Remarks to the Author):

In the present manuscript authors analyzed the role of the nuclear membrane protein, SAD1/UNC84 domain protein 2 (SUN2), which links cytoskeleton and nucleoskeleton, in flavivirus replication. Using CRISPR/Cas9-mediated knockout of SUN2, authors observed a significant reduction of ZIKV, DENV and JEV replication. Apparently, SUN2 participates in the replicative complex formation and promotes viral RNA synthesis. In a neonatal mouse infection model, SUN2 knockout alleviates the in vivo ZIKV replication and neuropathology.

The manuscript is very interesting and highlights the importance of the cytoskeleton and the interactions between nucleus and cytoplasm for the flavivirus replication complex formation. However, there are some aspects that would be important to analyze to support the conclusions of the work.

Major concerns

1. In the present work authors alter the expression of SUN1 and 2 proteins using CRISPR/Cas9 to knockout the expression of both proteins, or the transduction of lentivirus to induce the expression of the proteins. However, if SUN2 is required for flavivirus infection, is the expression of this protein modified during infection?
2. Is it possible to analyze the effect of the restoration of SUN2 in the assay using the ZIKV replicon?
3. Is Knockdown of SUN2 altering the ER morphology? Some changes are observed in Figure 3C.
4. Why is the staining of E protein in Figures 4 A, B and C in control infected cells different?
5. It should be useful to confirm the interaction between NS1 and actin by confocal microscopy in ctrl and SUN2ko infected cells.

Reviewer #3 (Remarks to the Author):

In this study, Huang and colleagues report the role of the host factor SUN2 in flavivirus replication. CRISPR-Cas9-generated SUN2 knock-out (KO) cells are much less permissive to flaviviruses than to other RNA viruses. The replication phenotypes are good and particularly convincing. The phenotypes observed with Zika virus (ZIKV) subgenomic replicon system unambiguously demonstrate that SUN2 regulate the viral RNA synthesis step of the life cycle. SUN2 knock-out neonatal mice are less susceptible to ZIKV than wt mice in a lethal ZIKV challenge in vivo, which demonstrates the physiological relevance of SUN2 in ZIKV pathogenesis. These strong data make this study interesting and relevant for the field, and further highlight the contribution of nuclear host factors in pathogenesis.

Based on co-immunoprecipitation assays using cells expressing individual ZIKV nonstructural proteins, the authors propose that SUN2 regulates flavivirus life cycle through the modulation of the interaction between viral NS1 and the actin cytoskeleton which is not reorganized by ZIKV in infected KO cells. This correlates with a decrease in the abundance of viral replication organelles in SUN2 KO cells. However, in my opinion, such conclusions are not supported by the data and too premature because of two main

reasons. First, with a decrease of several log₁₀ in viral replication in KO cells, it is expected that no viral organelle will be generated, and that the cytoskeleton will not be rearranged, simply because there are no viral proteins expressed (and hence no replication), and not the other way around. Any type of inhibition of replication (e.g., polymerase/protease inhibitor) would result in similar phenotype even without a direct role in cytoskeleton and endoplasmic reticulum (ER) dynamics. Given this chicken and egg situation, the authors must clearly demonstrate that SUN2 ER remodeling and cytoskeleton reorganization in a replication-independent system with equal amounts of viral proteins in both wt and KO cells. At that stage, no firm conclusion can be made since so far only based on the correlation between these morphological phenotypes and viral replication efficacy.

Second, and most importantly, I felt that there was a major topology paradox regarding the model proposed by the authors. NS1 is a soluble protein located inside the lumen of the ER and not a transmembrane protein as depicted in Figure 7. How is it possible then that it associates with the cytosolic actin located outside of the ER? Given that the authors used ionic detergents-containing RIPA buffer, the ER membranes must have been solubilized during cell lysis, rendering this otherwise impossible interaction possible. The authors must use other approaches to better decipher the possible link between NS1 and the cytoskeleton. In addition, I could not find any information about the cloning of the plasmids expressing the viral proteins in the Methods section. This is important to include. Did the authors clone NS1 with the N-terminal signal peptide located at the end of E coding sequence. Without such peptide, NS1 will not be inserted in the ER and would be cytosolic, which is not its proper localization. When observed in confocal microscopy, does NS1 expressed alone localized to the ER? Finally, the model involves a third compartment namely the nucleus and SUN2 is not in contact with either the ER lumen or the cytosol. How the authors envision such interplay between these compartments? Testing the involvement of the nesprins might be relevant to refine a mechanistic model.

Other comments:

- What is the impact of flaviviral infection on SUN2 expression?
- In fig 6A, it would be useful to show the quantification of the brain size along with magnified picture of this organ.
- I could not find any information regarding the source and reference about the ZIKV replicon system. This must be provided.
- In figure 6, was there a difference in the phenotypes depending on the sex of the infected mice?
- All the experiments were performed at 24 hours post-infection which is not the peak of replication and is reflected by the relatively low infectious titers in PFU assays (especially for DENV). Is the replication impairment in KO cells maintained at the 48 and 72 hours time points
- In figure 5A and extended figure 8A, NS2A and NS2B seem not to be expressed. Hence no conclusions can be drawn about these viral proteins and the presumed lack of interaction with cytoskeletal proteins.
- In Figure 5B, why is there a 27 kDa band in all vector controls? It does not seem to be non-specific

- What is the cellular localization of the overexpressed wt SUN2 (RES) in figure 2C.
- In the 6 hour time point of Figure 3B, it is hard for the viewer to evaluate the impact of the KO on the Rlu activity because of the log scale. An extra graph for this particular time point with a better resolved Y axis might help.
- Line 114: I could not find the JEV data which are written to be in extended Fig 3
- Considering that SUN2 functions are disrupted upon deletion and that SUN1 is not involved in viral replication, a SUN1-SUN2 swapping approach might be relevant to identify SUN2 determinants.
- Statements of lines 54, 55, 67 and 341 should be tuned down because the role of NS4B and NS2A in viral organelle biogenesis are not unambiguously demonstrated to my knowledge.

Point-by-point response letter

Reviewer #1 (Remarks to the Author):

In this manuscript, Huang et al. reported that the nuclear envelope protein, SUN2, plays a critical role in viral genome replication through its involvement in modulating the cytoskeleton and facilitating the formation of viral replication organelles. Initially, the authors demonstrated that cells lacking SUN2 exhibited a decrease in viral growth, which can be attributed to the impairment of viral replication organelle formation. Next, they showed that the F-actin fiber underwent re-organization around the viral replication site, which was found to be absent in cells lacking SUN2. Then, the authors demonstrated that NS1 interacts with beta-actin in a SUN2-dependent manner. Finally, in vivo experiments were conducted using a mouse model, wherein it was observed that the virulence of ZIKV was diminished in cells lacking SUN2. Overall, they clearly show the involvement of SUN2 in ZIKV genome replication. However, despite its significance, there is a dearth of research on the molecular mechanisms underlying SUN2's involvement in these processes.

Figure 1. To ensure the proper propagation of the virus, the author must demonstrate the impact of SUN2 knockout on cell growth, as the growth retardation of host cells can significantly impede viral propagation.

Answer: Thanks for the reviewer's helpful suggestion. We compared the growth curves of two SUN2^{KO} cell clones and control cells, and the data indicated that the SUN2 KO does not alter the cell growth (Fig. 1C), consistent with the observations that SUN2 knockout in mice did not affect the growth of mice in our study and previous report (*Proc Natl Acad Sci U S A. 2009*,

106(25):10207-12).

Figure 2. The imaging quality is inadequate, making it imperative to investigate the alterations in SUN2 localization following viral infection. Moreover, it is crucial to demonstrate changes in nesprin localization as well. Without such evidence, it would be difficult to discuss the role of SUN2 at the nuclear envelope in the formation of the viral replication organelle. To propose a mechanistic model for SUN2/cytoskeleton-mediated replication organelle formation, the author should conduct experiments using deletion mutants of SUN2.

Answer: We sincerely appreciate the reviewer's helpful comments.

We have supplemented more high-resolution imaging data accordingly. The IFM images showed that SUN2 was localized on the nuclear membrane of both mock- and ZIKV-infected control cells (Fig. 2C), while the SUN2 staining signal in the ZIKV-infected cells was much weaker, consistent with our western blot data (Fig. 1D). Therefore, the localization of SUN2 was not altered following viral infection, but its level was downregulated.

Furthermore, we carried out a set of assays to explore role of Nesprins in the ZIKV infection. First, co-IP data confirmed that Nesprin-1 is associated with SUN2, actin, and NS1 (Fig 7A). Secondly, we monitored the localization of Nesprin-1 by IFM assay. In the mock-infected cells, Nesprin-1 is distributed both on nuclear membrane and in the cytoplasm (Fig. 7B). In the ZIKV-infected control cells, the cytoplasmic part of Nesprin-1 was aggregated to the perinuclear region, implying that Nesprins are involved in the formation of replication organelle. In contrast, the redistribution of Nesprin-1 mediated by ZIKV was not observed in the SUN2^{KO} cells (Fig. 7B). Thirdly, expression of dominant negative (DN) forms of Nesprins significantly reduced the interaction between NS1 and actin, as well as viral replication levels (Fig. 7G and 7H). These findings provide support for our mechanistic model.

We performed replicon assay using three deletion mutants of SUN2 (SUN2^{ΔN+TM}, SUN2^{ΔCC}, and SUN2^{ΔSUN}) (Extended data Fig. 6A). Luciferase activities of replicon could not be restored in the SUN2 truncates-expressing cells, together with IFM data (Extended data Fig. 6B and 6C), demonstrating that each region is essential for SUN2 to mediate formation of replication organelles.

Figure 3. To assess alterations in the morphology of ER-derived replication organelles, it is imperative to provide fluorescent images depicting changes in the shape of the ER membrane upon virus infection, utilizing cells expressing fluorescent protein-fused ER markers.

Answer: Thanks for the reviewer's helpful suggestion. To monitor the change of ER membrane shape, we generated cells expressing mCherry-SEC61B (an ER marker) fusion protein, and performed IFM assay using anti-calnexin (another ER marker). The IFM images revealed that upon ZIKV infection, both of calnexin and SEC61B were aggregated around the nuclei in the control cells, but not in the SUN2^{KO} cells (Fig. 3F and 3G), indicating that SUN2 is required for the ER membrane remodeling induced by ZIKV.

Figure 4. A quantitative analysis is necessary to assess changes in cytoskeletal organization. They only presented single-cell images. Although the results presented in Fig.4E depict the entire viral life cycle, they do not specifically demonstrate viral genome replication. Hence, the author should conduct identical experiments using a replicon system.

Answer: Thanks the reviewer's helpful comment. We have added quantitative analysis of F-actin, tubulin, and vimentin gathered rate (Fig. 4D-F). We carried out replicon assay in the presence of three inhibitors (cytochalasin B, paclitaxel, and acrylamide). These inhibitors were able to reduce the luciferase activities of ZIKV WT replicon (Extended data Fig. 6D).

Figure 5. The experiments conducted do not reveal any interactions between the virus and SUN2. Therefore, to elucidate the molecular mechanism underlying SUN2's role in viral replication organelle formation, the authors must show the evidence indicating a physiological link between SUN2 and viral components.

Answer: Thank for reviewer's helpful suggestion. We have performed co-IP assay to detect the link between SUN2 and viral components. We found that SUN2 co-precipitated with skeleton proteins (actin, Nesprin-1) and ZIKV NS1 in the virus infection context (Fig. 7A), suggesting SUN2 is physiologically linked with viral NS1 through Nesprins and actin.

Reviewer #2 (Remarks to the Author):

In the present manuscript authors analyzed the role of the nuclear membrane protein, SADI/UNC84 domain protein 2 (SUN2), which links cytoskeleton and nucleoskeleton, in flavivirus replication. Using CRISPR/Cas9-mediated knockout of SUN2, authors observed a significant reduction of ZIKV, DENV and JEV replication. Apparently, SUN2 participates in the replicative complex formation and promotes viral RNA synthesis. In a neonatal mouse infection model, SUN2 knockout alleviates the in vivo ZIKV replication and neuropathology.

The manuscript is very interesting and highlights the importance of the cytoskeleton and the

interactions between nucleus and cytoplasm for the flavivirus replication complex formation. However, there are some aspects that would be important to analyze to support the conclusions of the work.

Major concerns

1. In the present work authors alter the expression of SUN1 and 2 proteins using CRISPR/Cas9 to knockout the expression of both proteins, or the transduction of lentivirus to induce the expression of the proteins. However, if SUN2 is required for flavivirus infection, is the expression of this protein modified during infection?

Answer: Thanks for the reviewer's helpful comment. We examined the SUN2 protein levels after ZIKV infection by western blot. The SUN2 protein level was gradually decreased upon infection (Fig. 1D), consistent with previous report (*Nature*, 2021, 596:558-564). We briefly tested whether SUN2 level is downregulated through ubiquitination- or lysosomal-mediated degradation using their inhibitors (MG132 and NH₄Cl). The treatment of these inhibitors did not rescue the downregulation of SUN2 (data not shown). The underlying mechanism that SUN2 protein is downregulated by viral infection will be investigated in future study.

2. Is it possible to analyze the effect of the restoration of SUN2 in the assay using the ZIKV replicon?

Answer: Thanks for the reviewer's helpful suggestion. We performed replicon assay using SUN2^{RES} cells, and found that luciferase activities of ZIKV WT replicon could be largely restored in the SUN2^{RES} cells (Fig. 3B, blue full-line).

3. Is Knockdown of SUN2 altering the ER morphology? Some changes are observed in Figure 3C.

Answer: We appreciate the reviewer's helpful suggestion. We generated control or SUN2 KO cells expressing mCherry-SEC61B (an ER marker), and performed IFM assay using anti-calnexin (another ER marker) to monitor the ER morphology. The IFM images showed that the ER membrane morphologies in the mock-infected control and SUN2 KO cells are similar. Upon ZIKV infection, calnexin and SEC61B were aggregated around the nuclei in the control cells, but not in the SUN2^{KO} cells, indicating that SUN2 is required for the ER membrane alteration induced by ZIKV (Fig. 3F and 3G).

4. Why is the staining of E protein in Figures 4 A, B and C in control infected cells different?

Answer: Thank for reviewer's comment. Based on our observations, the distribution patterns of E

protein might vary slightly in different cells depending on the image angles and its accumulation levels. In general, the E protein is predominantly located within the perinuclear region. In Fig. 4A-C, the E protein also surrounded nuclei, despite its distribution patterns were not completely identical.

5. It should be useful to confirm the interaction between NS1 and actin by confocal microscopy in ctrl and SUN2ko infected cells.

Answer: Thanks for reviewer's helpful suggestion. We performed the IFM assay to confirm the interaction between NS1 and actin. The images showed that in the ZIKV-infected control cells, NS1 co-localizes with F-actin, which was eliminated in the SUN2^{KO} cells (Fig. 5D), demonstrating that SUN2 is involved in the NS1-actin interaction.

Reviewer #3 (Remarks to the Author):

In this study, Huang and colleagues report the role of the host factor SUN2 in flavivirus replication. CRISPR-Cas9-generated SUN2 knock-out (KO) cells are much less permissive to flaviviruses than to other RNA viruses. The replication phenotypes are good and particularly convincing. The phenotypes observed with Zika virus (ZIKV) subgenomic replicon system unambiguously demonstrate that SUN2 regulate the viral RNA synthesis step of the life cycle. SUN2 knock-out neonatal mice are less susceptible to ZIKV than wt mice in a lethal ZIKV challenge in vivo, which demonstrates the physiological relevance of SUN2 in ZIKV pathogenesis. These strong data make this study interesting and relevant for the field, and further highlight the contribution of nuclear host factors in pathogenesis. Based on co-immunoprecipitation assays using cells expressing individual ZIKV nonstructural proteins, the authors propose that SUN2 regulates flavivirus life cycle through the modulation of the interaction between viral NS1 and the actin cytoskeleton which is not reorganized by ZIKV in infected KO cells. This correlates with a decrease in the abundance of viral replication organelles in SUN2 KO cells. However, in my opinion, such conclusions are not supported by the data and too premature because of two main reasons.

First, with a decrease of several log10 in viral replication in KO cells, it is expected that no viral organelle will be generated, and that the cytoskeleton will not be rearranged, simply because there are no viral proteins expressed (and hence no replication), and not the other way around. Any type of inhibition of replication (e.g., polymerase/protease inhibitor) would result in similar phenotype even without a direct role in cytoskeleton and endoplasmic reticulum (ER) dynamics. Given this chicken and egg situation, the authors must clearly demonstrate that SUN2 ER remodeling and

cytoskeleton reorganization in a replication-independent system with equal amounts of viral proteins in both wt and KO cells. At that stage, no firm conclusion can be made since so far only based on the correlation between these morphological phenotypes and viral replication efficacy.

Answer: Thank you for your helpful comments. To avoid the chicken and egg situation, we ectopically expressed equal amounts of ZIKV NS1 in the control and SUN2 KO cells by transfection, rather than ZIKV infection, and compared the ER and cytoskeleton remodeling by IFM. The IFM imaging data showed that expression of ZIKV NS1 alone was sufficient to induce remodeling of ER and actin into the perinuclear region in the control cells (Fig. 6B), consistent with previous reports (*J Cell Biol*, 2020, 219(2): e201903062; *Virol J*, 2013, 10:339). Importantly, SUN2 KO significantly reduced the rearrangement of ER and actin mediated by NS1 (Fig. 6B), suggesting an involvement of SUN2 in the NS1-mediated cytoskeleton reorganization and ER remodeling.

Second, and most importantly, I felt that there was a major topology paradox regarding the model proposed by the authors. NS1 is a soluble protein located inside the lumen of the ER and not a transmembrane protein as depicted in Figure 7. How is it possible then that it associates with the cytosolic actin located outside of the ER? Given that the authors used ionic detergents-containing RIPA buffer, the ER membranes must have been solubilized during cell lysis, rendering this otherwise impossible interaction possible. The authors must use other approaches to better decipher the possible link between NS1 and the cytoskeleton. In addition, I could not find any information about the cloning of the plasmids expressing the viral proteins in the Methods section. This is important to include. Did the authors clone NS1 with the N-terminal signal peptide located at the end of E coding sequence. Without such peptide, NS1 will not be inserted in the ER and would be cytosolic, which is not its proper localization. When observed in confocal microscopy, does NS1 expressed alone localized to the ER? Finally, the model involves a third compartment namely the nucleus and SUN2 is not in contact with either the ER lumen or the cytosol. How the authors envision such interplay between these compartments? Testing the involvement of the nesprins might be relevant to refine a mechanistic model.

Answer: We sincerely appreciate reviewer's helpful comments.

Yes, NS1 is initially located in the lumen of ER upon proprotein cleavage. Later on, it can be transported to other compartments including viral replication sites to support viral RNA replication and virion production (*PLoS Pathog*, 2015, 11(11):e1005277). Theoretically, it is possible for NS1 to interact with cytoplasmic actin. In addition, our IFM images showed that in the ZIKV-infected control cells, NS1 protein was co-localized with SEC61B in the perinuclear region (Fig. 6A), and

SUN2 KO alleviated the gathered degree of NS1 and SEC61B proteins (Fig. 6A). In addition, AlphaFold2 prediction of NS1 and actin interaction suggested several potential interaction interfaces between NS1 and actin (Extended Data Figure 10). Therefore, we proposed that NS1 could be transported to the cytoplasm from ER lumen, and aid to form the replication organelles through interacting actin (Fig. 5B-5D).

We have added the info of ZIKV proteins in the Materials and Methods section (page 20, lines 565-571). Our construct expressing NS1 does not include the end of E coding sequence, so the ectopically-expressed NS1 might mainly reside in cytoplasm as reviewer pointed out. However, we have further performed co-IP and IFM assays to demonstrate the association between actin and viral NS1 in the context of viral infection. The co-IP data confirmed an association between actin and NS1, together with Nesprin-1 and SUN2 (Fig. 7A). IFM images also showed a colocalization of actin and NS1 (Fig. 5D), and NS1 protein was co-localized with SEC61B in the perinuclear region of control cells (Fig. 6A). These observations suggested that NS1 can be localized to ER, and interact with cytoskeleton actin.

To test involvement of Nesprins in the association between SUN2 and actin and in the ZIKV replication, we performed following assays: (1), Co-IP data confirmed that Nesprin-1 is associated with SUN2, actin, and NS1 in the context of ZIKV infection (Fig 7A). (2), IFM data showed that upon ZIKV infection, the cytoplasmic part of Nesprin-1 became aggregated to the perinuclear region and colocalized with ZIKV E protein in the control cells, which was impaired in the cytoplasm of SUN2^{KO} cells (Fig 7B). (3), Expression of dominant negative (DN) forms of Nesprins dramatically alleviated the NS1-actin interaction (Fig. 7F) and viral replication levels (Fig. 7G and 7H). These collective data demonstrated that Nesprins play a role in the NS1-actin association and participate in the viral replication, providing support for our mechanistic model.

Other comments:

- What is the impact of flaviviral infection on SUN2 expression?

Answer: Thanks for the reviewer's comment. In response to ZIKV infection, the SUN2 protein levels in Huh7 cells were gradually decreased (Fig. 1D), consistent with previous report (*Nature*, 2021, 596:558-564). We have briefly tested whether SUN2 level is downregulated through ubiquitination-mediated or lysosomal-mediated degradation using their inhibitors (MG132 and NH₄Cl). The treatment of these inhibitors did not rescue the downregulation of SUN2 (data not shown). The underlying mechanism that SUN2 protein is downregulated by viral infection will be investigated in future study.

- I could not find any information regarding the source and reference about the ZIKV replicon system. This must be provided.

Answer: We have added the source and reference of the ZIKV replicon system in the revised manuscript as suggested (page 23, lines 629-630).

- In figure 6, was there a difference in the phenotypes depending on the sex of the infected mice?

Answer: Because the immune system of neonatal mice is not completely developed and the growth rates are comparable in male and female mice, we did not pay much attention to the impact of gender on the pathogenesis of mice. As we did not specifically distinguish genders of tested mice, we did not know if there is a difference in the phenotypes.

- All the experiments were performed at 24 hours post-infection which is not the peak of replication and is reflected by the relatively low infectious titers in PFU assays (especially for DENV). Is the replication impairment in KO cells maintained at the 48 and 72 hours time points.

Answer: Thanks for your helpful suggestion. We performed multi-step growth assay, and the data showed that SUN2 KO led to dramatic reductions of ZIKV yields at 24, 48, and 72 h p.i. (Fig. 1H).

- In figure 5A and extended figure 8A, NS2A and NS2B seem not to be expressed. Hence no conclusions can be drawn about these viral proteins and the presumed lack of interaction with cytoskeletal proteins.

Answer: Yes, expression of NS2A and NS2B is challenging. They could be only detected upon co-immunoprecipitation (Extended data Fig. 8A). Based on current data, we could not draw a firm conclusion about the interaction between NS2A or NS2B proteins and cytoskeletal proteins, so we modified our conclusion accordingly (lines 266-268).

- In Figure 5B, why is there a 27 kDa band in all vector controls? It does not seem to be non-specific

Answer: In Fig. 5B, the 27 kDa band is HA-GFP protein, which is an irrelevant protein set as control. We have corrected the misleading info in Fig. 5B and figure legend (lines 976-977).

- What is the cellular localization of the overexpressed wt SUN2 (RES) in figure 2C.

Answer: The cellular localization of the overexpressed SUN2 (RES) is distributed on the nuclear membrane, like endogenous SUN2 (Fig. 2C).

- In the 6 hour time point of Figure 3B, it is hard for the viewer to evaluate the impact of the KO on the Rlu activity because of the log scale. An extra graph for this particular time point with a better resolved Y axis might help.

Answer: Thanks for your helpful suggestion. We have repeated the replicon assay, together with SUN^{RES} cells, and replotted the graph (Fig. 3B). In the revised figure, the impact of SUN2 KO on the replicon activity should be easy to evaluate.

- Line 114: I could not find the JEV data which are written to be in extended

Fig 3 Answer: The JEV data were presented in **Extended** data Fig. 3, not in the Fig. 3.

- Considering that SUN2 functions are disrupted upon deletion and that SUN1 is not involved in viral replication, a SUN1-SUN2 swapping approach might be relevant to identify SUN2 determinants.

Answer: Thanks for the reviewer's helpful suggestion. As only SUN2 plays an essential role in the viral replication, identification of SUN2 determinants will be definitely helpful to illustrate the differences between SUN2 and SUN1. We will take your suggestion to swap SUN1 and SUN2 regions in near future, and include the data in next manuscript.

- Statements of lines 54, 55, 67 and 341 should be tuned down because the role of NS4B and NS2A in viral organelle biogenesis are not unambiguously demonstrated to my knowledge.

Answer: Thanks for reviewer's helpful suggestion. We have carefully revised these lines accordingly (lines 61, 62, 74, 466 in revised manuscript).

In summary, we have addressed the comments of reviewers in full, and carefully revised our manuscript. A copy of revised manuscript with changes in red has been uploaded to the system. We hope that with incorporation of the suggested revisions, the manuscript will be judged acceptable for publication in *Nature Communications*. Thanks for your consideration!

Sincerely yours,
Ping Zhang

REVIEWERS' COMMENTS

Reviewer #1 (Remarks to the Author):

The authors appear to have addressed all of the raised concerns, and as a result, the manuscript now appears suitable for publication in Nature Communications.

Reviewer #2 (Remarks to the Author):

The manuscript is now suitable for publication

Reviewer #3 (Remarks to the Author):

In this revised version of the manuscript, the authors have successfully addressed most of my concerns with a substantial number of experiments which represents a lot of work. Thus, I now recommend the publication of this convincing study into Nature Communications. That said, I do not agree with the interpretation of the authors in their answer regarding the topology of NS1. I do not believe that a potential cytoplasmic localization of NS1 is supported by any published study so far. NS1 is always located in the ER lumen and the interior of a replication vesicle is cytosolic content. Although I accept the manuscript, I would strongly recommend to the authors to reconsider some conceptual statement about NS1 topology and interactions with cytoplasmic components such as Nesprins. This is most likely indirect and mediated by other proteins functionally connecting the lumen of the ER with the cytoplasm.

Point-by-point response letter

Reviewer #1 (Remarks to the Author):

The authors appear to have addressed all of the raised concerns, and as a result, the manuscript now appears suitable for publication in Nature Communications.

Reviewer #2 (Remarks to the Author):

The manuscript is now suitable for publication

Reviewer #3 (Remarks to the Author):

In this revised version of the manuscript, the authors have successfully addressed most of my concerns with a substantial number of experiments which represents a lot of work. Thus, I now recommend the publication of this convincing study into Nature Communications. That said, I do not agree with the interpretation of the authors in their answer regarding the topology of NS1. I do not believe that a potential cytoplasmic localization of NS1 is supported by any published study so far. NS1 is always located in the ER lumen and the interior of a replication vesicle is cytosolic content. Although I accept the manuscript, I would strongly recommend to the authors to reconsider some conceptual statement about NS1 topology and interactions with cytoplasmic components such as Nesprins. This is most likely indirect and mediated by other proteins functionally connecting the lumen of the ER with the cytoplasm.

A: Thanks for reviewer's suggestion. We have rephrased our statement accordingly (lines 443-448), which were highlighted in red.

Sincerely yours,

Ping Zhang